# SMALL MOLECULE RETRIEVAL FROM TANDEM MASS SPECTROMETRY: WHAT ARE WE OPTIMIZING FOR?

## ABSTRACT

One of the central challenges in the computational analysis of liquid chromatography-tandem mass spectrometry (LC-MS/MS) data is to identify the compounds underlying the output spectra. In recent years, this problem is increasingly tackled using deep learning methods. A common strategy involves predicting a molecular fingerprint vector from an input mass spectrum, which is then used to search for matches in a chemical compound database. While various loss functions are employed in training these predictive models, their impact on model performance remains poorly understood. Facilitated by the recent establishment of standardized benchmarks, we investigate in this study commonly used loss functions through both an empirical and theoretical lens. Our results reveal a fundamental trade-off between the two objectives of (1) fingerprint similarity and (2) molecular retrieval. In conclusion, optimizing for more accurate fingerprint predictions typically translates into worse retrieval results, and vice versa.

## 1 INTRODUCTION

Liquid chromatography-tandem mass spectrometry (LC-MS/MS) is a standard analytical approach applied towards the identification of small molecules in biological samples (i.e., metabolites) (de Jonge et al., 2022). The technique first separates compounds (LC), after which they are ionized and measured based on mass ($MS^1$). Finally, the ions are fragmented ($MS^2$). As a result, each selected precursor ion (i.e., molecule) in a sample produces a spectrum of fragments, each represented by a peak of a certain intensity and mass over charge (*m/z*). The characterization of an unknown compound is, hence, based on the fragmentation patterns of its ions. One of the central ideas of computational mass spectrometry is to develop methods to characterize the molecular compound underlying an $MS^2$ spectrum (Stravs, 2024). This task is complicated due to, among others: (1) the complexity and diversity of chemical structures, (2) the variability of fragmentation behavior, and (3) limitations of reference spectral databases.

From a computational perspective, compound identification approaches fall into one of three categories (Butler et al., 2023). The first category entails comparing the obtained fragmentation mass spectrum to a reference database of molecule-spectrum pairs, commonly called *spectral library matching* (Huber et al., 2021). The second type of method involves predicting some molecular property (vector), which can then be used to index into a molecular database, commonly called *molecular retrieval* (Dührkop et al., 2015). The third and final approach is to predict the molecular structure directly, using some structured output or generative modeling approach. This last method is commonly called *de novo structure elucidation* (Stravs et al., 2022; Butler et al., 2023; Bohde et al., 2025).

In this study, the focus is on retrieving compounds from a molecular database. The paradigmatic machine learning approach to this problem is to predict molecular fingerprints—(typically) sparse binary vector representations encoding the presence of substructures or chemical features (Cereto-Massagué et al., 2015). Fingerprints are constructs commonly used to compare molecular structures via similarity functions such as the Tanimoto or cosine similarity (Bajusz et al., 2015).

There have been recent efforts in compiling datasets of paired spectrum–molecule (supervised) data, such as (1) the GNPS repository (Wang et al., 2016), and (2) the more recent machine learning-focused MassSpecGym benchmark (Bushuiev et al., 2024a). Spurred on by these developments, many recent works have proposed to tackle molecular retrieval with deep learning

methodologies. Despite growing interest in this area, there is a limited understanding of how best to predict fingerprints, or—in other words—how the choice of loss function affects the final model.

In the present study, we conduct a systematic evaluation of loss functions for training models that predict molecular fingerprint vectors. Given the practical use of these vectors towards molecular identification, we investigate the performance of these loss functions w.r.t. both fingerprint accuracy (in terms of Tanimoto similarities) and retrieval performance (in terms of average hit rate @ $k$). Our experiments reveal an apparent paradox: models that predict more accurate fingerprints do not achieve better retrieval results (Figure 1). In other words: the "best" fingerprint prediction is not necessarily the most discriminative one. To explain this phenomenon, we conduct a theoretical analysis for Bayes-optimal decisions of the different metrics each loss is optimizing for, and define novel regret bounds.

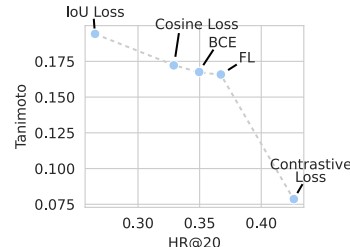

Figure 1: Empirically, different loss functions form a pareto front trading off fingerprint similarity (y-axis) and retrieval performance (x-axis). See Section 3 for experimental details. Remark that maximizing Tanimoto similarity is the same as minimizing IoU loss.

## 2 PROBLEM SETTING

In this section, we first define the prediction setting and the metrics of interest (Section 2.1). Afterwards, commonly used loss functions to predict fingerprints are discussed (Section 2.2).

### 2.1 PREDICTING FINGERPRINTS

Let the input domain that consists of mass spectra be denoted by $\mathcal{X}$. Let the output domain that consists of molecules be denoted by $\mathcal{Y}$. Here, the output domain of molecules constitutes of all chemically feasible $m$-dimensional molecular fingerprints (i.e., $\mathcal{Y} \subset \{0,1\}^m$). Let us denote a dataset of mass spectrum–molecule pairs by $\mathcal{D} = \{(\boldsymbol{x}^{(i)}, \boldsymbol{y}^{(i)})\}_{i=1}^n$, where each $\boldsymbol{x}^{(i)} \in \mathcal{X}$ and $\boldsymbol{y}^{(i)} \in \mathcal{Y}$ denote an input mass spectrum and the corresponding fingerprint of the ground truth molecule. For each input $i$, one can define a set of retrieval candidate fingerprints $\mathcal{C}^{(i)} = \{\boldsymbol{c}_1^{(i)}, \ldots, \boldsymbol{c}_l^{(i)}\}$, with $\boldsymbol{c}_j^{(i)} \in \{0,1\}^m$ for all $j \in \{1, \ldots, l\}$. Retrieval candidates can be drawn from a database of chemicals based on prior information on the sample. For example, candidate molecules can be defined as (1) the subset of known structures with equal mass to the molecule underlying the spectrum, or (2) the subset of known structures sharing the same molecular formula to the true molecule. The former is (typically) known through MS[1] information, while the latter can usually be obtained by spectral processing pipelines (Dührkop et al., 2019). By definition, the ground truth fingerprint $\boldsymbol{y}^{(i)}$ should be part of the candidate set $\mathcal{C}^{(i)}$.

The general goal of this study is to formulate deep learning models that predict fingerprints $\hat{\boldsymbol{y}}^{(i)} \in [0,1]^m$. Let us denote the fingerprint predictor as $f$, producing fingerprint predictions according to:

$$\hat{\boldsymbol{y}}^{(i)} = f(\boldsymbol{x}^{(i)}) = \sigma(\boldsymbol{W}_{\text{out}} E(\boldsymbol{x}^{(i)})),$$

where all but the last layers are included in the embedder $E$, which can be any neural network function embedding the spectra to a $d$-dimensional space. A last linear layer, parameterized by a learnable $\boldsymbol{W}_{\text{out}} \in \mathbb{R}^{m \times d}$, then projects the embedding to fingerprint space, upon which a sigmoid operation $\sigma$ is applied. An overview of this predictive framework is visualized in Figure 2A.

The performance of fingerprint prediction models is usually assessed through both (1) evaluating the accuracy of predicted fingerprints, and (2) evaluating their performance in retrieving the ground truth molecule from a set of candidate fingerprints (Bushuiev et al., 2024a; Goldman et al., 2023). Fingerprint accuracy is usually assessed through the Tanimoto similarity (Bajusz et al., 2015), a metric also known as the Jaccard index and the intersection-over-union (IoU):

$$U_{\text{Tan}}(\boldsymbol{y}^{(i)}, \hat{\boldsymbol{y}}^{(i)}) = \frac{\hat{\boldsymbol{y}}^{(i)\top} \boldsymbol{y}^{(i)}}{\|\hat{\boldsymbol{y}}^{(i)}\|_1 + \|\boldsymbol{y}^{(i)}\|_1 - \hat{\boldsymbol{y}}^{(i)\top} \boldsymbol{y}^{(i)}},$$

**(A)** Molecular retrieval based on fingerprint prediction

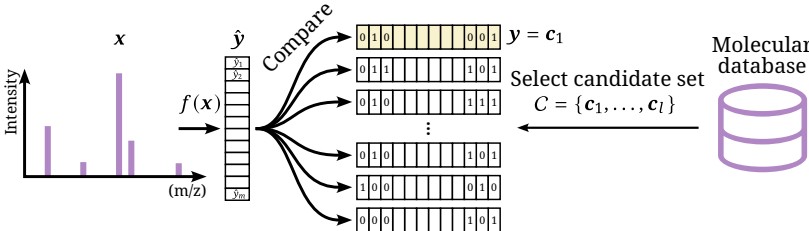

**(B)** Loss functions for fingerprint prediction

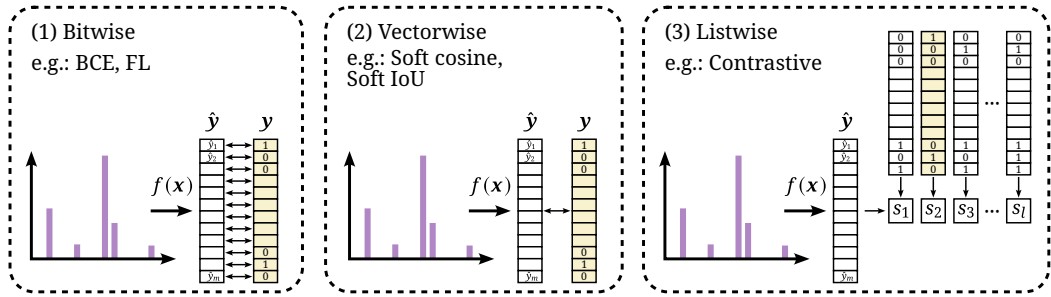

Figure 2: **(A)** General framework used in this study. In this figure, all superscript notations indicating the sample index $(i)$ are omitted. A model $f$ predicts a fingerprint $\hat{y}^{(i)} \in [0,1]^m$ from an input spectrum $x^{(i)}$. The prediction is then compared to a set of fingerprints $\mathcal{C}^{(i)}$ derived from candidate molecules in a database. The true fingerprint $y^{(i)}$ is assumed to be part of the candidate set. **(B)** Loss functions for training $f(\cdot)$ generally fall into one of three categories: (1) bitwise losses (e.g., binary cross entropy or binary focal loss), (2) vectorwise losses (e.g., soft cosine loss or soft IoU loss), or (3) listwise losses (e.g., contrastive loss).

where $\|\cdot\|_1$ denotes the L1 norm. While the formula does not strictly require it, Tanimoto similarities are usually computed over two binary vectors. To this end, predictions are (usually) first binarized by defining a threshold (usually 0.5) (Goldman et al., 2023).

To assess retrieval performance, the MassSpecGym benchmark employs the hit rate @ $k$ (Bushuiev et al., 2024a). To compute this score, a set of matching scores $\mathcal{S}^{(i)}$ is first computed:

$$\mathcal{S}^{(i)} = \{s_1^{(i)}, \ldots, s_l^{(i)}\}, \quad \text{where } s_j^{(i)} = \text{sim}(c_j^{(i)}, \hat{y}^{(i)}) \quad \forall j \in \{1, \ldots, l\},$$

with $\text{sim}(\cdot, \cdot)$ usually the Tanimoto or cosine similarity. This set quantifies a "matching score" of the predicted fingerprint against all the candidates in the set $\mathcal{C}^{(i)}$. Let $s_{y^{(i)}}$ denote the matching score to the true molecule $y^{(i)}$. Then, let us define the set of $k$ highest scores as $\mathcal{S}_k^{(i)}$, where $\mathcal{S}_k^{(i)} \subseteq \mathcal{S}^{(i)}$. For a test instance indexed by $i$ the hit rate @ $k$ utility is then computed as:

$$U_{\text{HR@}k}(y^{(i)}, \mathcal{S}_k^{(i)}) = \mathbb{I}\left[s_{y^{(i)}} \in \mathcal{S}_k^{(i)}\right],$$

where $\mathbb{I}$ is the indicator function.

### 2.2 LOSS FUNCTIONS FOR FINGERPRINT PREDICTION

The central idea of this study is to investigate how to define learning objectives to maximize performance w.r.t. the metrics in the previous section. Broadly speaking, we can categorize the loss functions previously employed into one of three categories, here called: (1) bitwise, (2) vectorwise, and (3) listwise. All three categories are visualized in Figure 2B. In the following, all loss functions are assumed to receive soft predictions $\hat{y}^{(i)} \in [0,1]^m$.

**Bitwise losses.** The simplest way to approach the problem of fingerprint prediction is through the lens of multi-label classification. For deep learning, multi-label loss functions are usually label-wise

decomposable (Dembczyński et al., 2012). The archetypical loss in this category is the binary cross entropy (BCE), defined for a single sample $(i)$ as:

$$\text{BCE}(\boldsymbol{y}^{(i)}, \hat{\boldsymbol{y}}^{(i)}) = -\sum_{k=1}^{m} y_k^{(i)} \log(\hat{y}_k^{(i)}) + (1 - y_k^{(i)}) \log(1 - \hat{y}_k^{(i)}). \tag{1}$$

Hence, in this study, we take "bitwise losses" to mean all losses that seek to optimize the predictions of individual bits using label-wise decomposable functions. The focal loss (FL) (Lin et al., 2017) is a modification of the BCE that down-weights easy examples so the model focuses more on learning from misclassified examples:

$$\text{FL}(\boldsymbol{y}^{(i)}, \hat{\boldsymbol{y}}^{(i)}) = -\sum_{k=1}^{m} y_k^{(i)} (1 - \hat{y}_k^{(i)})^\gamma \log(\hat{y}_k^{(i)}) + (1 - y_k^{(i)})(\hat{y}_k^{(i)})^\gamma \log(1 - \hat{y}_k^{(i)}), \tag{2}$$

with $\gamma$ a hyperparameter. As it can reduce the contribution of the majority class to the total loss, the focal loss arises as a natural drop-in replacement for BCE in the context of imbalanced label problems. Molecular fingerprint label vectors are sparse. On average, a molecule in the training set only counts $\pm 46$ positive bits out of 4096 in total.

**Vectorwise losses.** As a second category, we consider loss functions that optimize the entire predicted label vector at once—or, in other words, are not label-wise decomposable. Molecular fingerprint predictions are commonly optimized using the soft cosine similarity loss (Goldman et al., 2023; Bushuiev et al., 2024a;b):

$$\text{CosineLoss}(\boldsymbol{y}^{(i)}, \hat{\boldsymbol{y}}^{(i)}) = 1 - \frac{\hat{\boldsymbol{y}}^{(i)\top} \boldsymbol{y}^{(i)}}{\|\hat{\boldsymbol{y}}^{(i)}\|_2 \cdot \|\boldsymbol{y}^{(i)}\|_2}. \tag{3}$$

Alternatively, the Tanimoto similarity can be directly maximized (Nowozin, 2014; Wang et al., 2023). This is equivalent to minimizing the Tanimoto-loss or IoU loss[1].

**Listwise losses.** A last category of loss functions attempts to learn how an input spectrum relates to every putative structure in the candidate list $\mathcal{C}^{(i)}$, here referred to as "listwise" losses. A prevalent example in this category are the family of contrastive losses (Chen et al., 2020). In metabolomics, they are usually formulated along the line of:

$$\text{ContrastiveLoss}(\boldsymbol{y}^{(i)}, \mathcal{C}^{(i)}) = -\log \frac{\exp\left(g\left(\boldsymbol{x}^{(i)}, \boldsymbol{y}^{(i)}\right)/\tau\right)}{\sum_{\boldsymbol{c}_j^{(i)} \in \mathcal{C}^{(i)}} \exp\left(g\left(\boldsymbol{x}^{(i)}, \boldsymbol{c}_j^{(i)}\right)/\tau\right)}, \tag{4}$$

where $\boldsymbol{y}^{(i)}$ is the true candidate, $\tau$ is a temperature hyperparameter, and $g$ is any function that returns a scalar score quantifying the match between spectrum (embedding) and candidate fingerprints (Goldman et al., 2023; Kalia et al., 2025; Chen et al., 2024). Remark that that the above loss function essentially converts the multi-label fingerprint prediction problem into a multi-class classification problem, where the singular positive class is the correct fingerprint and all other candidate label bitvectors are taken as negative classes. As it is computationally infeasible to enumerate a softmax over all possible label vectors $\boldsymbol{y}^{(i)} \in \{0,1\}^m$, the usage of the candidate set $\mathcal{C}^{(i)}$, with $|\mathcal{C}^{(i)}| \ll 2^m$, can be seen as some form of deterministic negative sampling (Mikolov et al., 2013).

In this study we evaluate different formulations of $g$. A first distinction can be made between those that learn a matching score using (the space of) fingerprints directly, or using a learned projection of the fingerprints. For instance, $g$ can be formulated using the cosine similarity function, either on the predicted and candidate fingerprints directly, or on learned (lower-dimensional) projections thereof. In the latter case, an extra linear layer $\boldsymbol{W}_{\text{spec}} \in \mathbb{R}^{h \times d}$ can be used to project the $d$-dimensional embedding space of the penultimate layer to a lower $h$-dimensional on which cosine similarities are computed (in this study, dimensions are chosen as $d = 1024$, $h = 256$). To project candidate fingerprints to the same space, a separate learned linear projection $\boldsymbol{W}_{\text{cand}} \in \mathbb{R}^{h \times m}$ can be used. A second distinction can be made in terms of the function used to obtain the final matching score. This

---

[1]In this paper we will use the notion Tanimoto-similarity when maximizing a utility, and the IoU loss when minimizing a loss.

function can be either fixed (e.g., the cosine similarity), or learned (e.g., using an MLP). Together, these choices compose the four different options for $g$ evaluated in this study:

$$g_{\text{Fp-Cos}}\left(\boldsymbol{x}^{(i)}, \boldsymbol{c}_j^{(i)}\right) = \cos\left(f(\boldsymbol{x}^{(i)}), \boldsymbol{c}_j^{(i)}\right), \tag{5}$$

$$g_{\text{Emb-Cos}}\left(\boldsymbol{x}^{(i)}, \boldsymbol{c}_j^{(i)}\right) = \cos\left(\boldsymbol{W}_{\text{spec}} E(\boldsymbol{x}^{(i)}), \boldsymbol{W}_{\text{cand}} \boldsymbol{c}_j^{(i)}\right), \tag{6}$$

$$g_{\text{Fp-MLP}}\left(\boldsymbol{x}^{(i)}, \boldsymbol{c}_j^{(i)}\right) = \text{MLP}\left(f(\boldsymbol{x}^{(i)}), \boldsymbol{c}_j^{(i)}\right), \tag{7}$$

$$g_{\text{Emb-MLP}}\left(\boldsymbol{x}^{(i)}, \boldsymbol{c}_j^{(i)}\right) = \text{MLP}\left(\boldsymbol{W}_{\text{spec}} E(\boldsymbol{x}^{(i)}), \boldsymbol{W}_{\text{cand}} \boldsymbol{c}_j^{(i)}\right). \tag{8}$$

In existing works, the formulation using cosine similarity on a lower-dimensional space ("Emb-Cos") is most popular (Goldman et al., 2023; Kalia et al., 2025). In the case that the contrastive loss operates on the penultimate layer of the fingerprint predictor ("Emb" models), the last layer is not utilized and, hence, removed from the model. For more details on the exact configurations of the MLP above, as well as the neural network embedder $E(\boldsymbol{x}^{(i)})$, see Appendix A. In addition, for an extended comparison of our setup with prior work, see Appendix B.

## 3 EMPIRICAL RESULTS

In this section we showcase that a trade-off between retrieval performance and fingerprint accuracy is observed empirically.

**Experimental setup.** To train and test models, we use the recently published MassSpecGym benchmark (Bushuiev et al., 2024a). MassSpecGym consists of 231 104 spectra–molecule pairs. In total, 28 929 unique molecules are present in MassSpecGym.[2] MassSpecGym allocates all spectra corresponding to a molecule either to a training, validation, or test split. The assignment of molecules to a split is decided upon by clustering them according to their Maximum Common Edge Subgraph (MCES) distance (Kretschmer et al., 2025). The splitting methodology ensures no molecules from the same cluster are found in different data splits. Hence, MassSpecGym ships with a generalization-demanding data split. The training, validation, and test set contain 194 119, 19 429, and 17 556 spectra, each covering 22 746, 3 185, and 2 998 molecules, respectively.

For every molecule in MassSpecGym, the true molecule is represented by a 4096-dimensional Morgan fingerprint (Rogers & Hahn, 2010) (computed using a radius of 2). Multiple ways exist to define retrieval candidate sets $\mathcal{C}^{(i)}$ from a database of chemical compounds. MassSpecGym uses the two retrieval settings explained above: candidates drawn from the set of known molecules with (1) equal mass, or (2) equal chemical formula. For both settings, separate retrieval metrics are computed and reported. In the benchmark, retrieval candidates are drawn from a database of chemical compounds up to a max of $|\mathcal{C}^{(i)}| \leq 256$, for all $i \in \{1, \ldots, l\}$.

As explained in Section 2.1, fingerprint predictions are produced according to $\hat{\boldsymbol{y}}^{(i)} = f(\boldsymbol{x}^{(i)}) = \sigma(\boldsymbol{W}_{\text{out}} E(\boldsymbol{x}^{(i)})) \in [0, 1]^{4096}$. In this study, the spectrum embedder $E$ is an MLP operating on binned input spectra $\boldsymbol{x}^{(i)}$ bounded by $m/z$ values in the interval $[0, 1005]$, with a binning width of 0.1. Hence, the dimensionality of input spectra $\boldsymbol{x}^{(i)} \in \mathbb{R}^f$ is $f = 10050$. Architectural details are given in Appendix A. To optimize the MLPs, we use one of the eight loss functions explained in Section 2.2: (1) BCE, (2) FL, (3) Cosine Loss, (4) IoU Loss, (5) Contrastive FP-Cos, (6) Contrastive Emb-Cos, (7) Contrastive Fp-MLP, and (8) Contrastive Emb-MLP. For every loss function, the learning rate is tuned separately. Test results are then shown using the models with the optimal learning rate, validation-wise. For the focal and all contrastive losses, the additional hyperparameters of $\gamma$ and $\tau$, respectively, are additionally tuned. Model training and tuning details are given in Appendix A.

Model performance is evaluated using (1) the average Tanimoto similarity of the prediction to the fingerprint of the true molecule, and (2) the average hit rate @ $k$ using different retrieval candidate settings (see Section 2.1). Note that, for computing validation and test Tanimoto similarities, all pre-

---

[2] On average, $\pm 8$ different spectra are measured per molecule. The distribution of spectra per molecule is considerably left-tailed, however, as the median number of spectra per molecule equals $\pm 3$. The maximum number of input spectra for a single molecule is 548.

Table 1: Molecular retrieval and fingerprint accuracy performance on MassSpecGym (Bushuiev et al., 2024a). The best and second-best performing models for each metric are indicated with boldface and underlined, respectively. For the models trained in this study, numbers report averages and standard deviations over 5 model runs. The performances of a random model and the MIST model (Goldman et al., 2023) are taken from Bushuiev et al. (2024a).

| | Retrieval w. equal mass candidates | | | Retrieval w. equal formula candidates | | | |
| | HR@1 | HR@5 | HR@20 | HR@1 | HR@5 | HR@20 | Tanimoto (IoU) |
|---|---|---|---|---|---|---|---|
| **This study** | | | | | | | |
| BCE | $10.12 \pm 0.36$ | $19.44 \pm 0.16$ | $34.96 \pm 0.73$ | $10.08 \pm 0.22$ | $22.60 \pm 0.35$ | $40.79 \pm 0.41$ | $16.75 \pm 0.11$ |
| FL | $11.54 \pm 0.14$ | $20.84 \pm 0.27$ | $36.72 \pm 0.44$ | $11.15 \pm 0.14$ | $23.82 \pm 0.23$ | $42.76 \pm 0.53$ | $16.58 \pm 0.16$ |
| Cosine Loss | $09.48 \pm 0.19$ | $18.19 \pm 0.17$ | $32.90 \pm 0.38$ | $09.12 \pm 0.41$ | $21.33 \pm 0.30$ | $39.38 \pm 0.31$ | $\underline{17.22 \pm 0.09}$ |
| IoU Loss | $05.58 \pm 0.32$ | $13.01 \pm 0.39$ | $26.51 \pm 0.50$ | $06.48 \pm 0.32$ | $17.62 \pm 0.51$ | $35.05 \pm 0.22$ | $\mathbf{19.42 \pm 0.13}$ |
| Contrastive Fp-Cos | $\underline{12.30 \pm 0.67}$ | $\mathbf{26.38 \pm 0.95}$ | $42.65 \pm 2.18$ | $11.37 \pm 0.31$ | $25.28 \pm 0.36$ | $\underline{45.04 \pm 0.76}$ | $07.86 \pm 0.99$[1] |
| Contrastive Emb-Cos | $12.29 \pm 0.69$ | $\underline{26.09 \pm 1.43}$ | $\underline{45.05 \pm 0.71}$ | $\mathbf{13.62 \pm 1.12}$ | $\mathbf{26.33 \pm 1.06}$ | $\mathbf{46.50 \pm 0.94}$ | N/A |
| Contrastive Fp-MLP | $09.72 \pm 0.20$ | $22.99 \pm 1.80$ | $42.34 \pm 1.08$ | $12.28 \pm 0.79$ | $24.34 \pm 0.37$ | $43.49 \pm 0.80$ | N/A |
| Contrastive Emb-MLP | $11.08 \pm 0.40$ | $25.66 \pm 0.82$ | $43.91 \pm 2.23$ | $\underline{12.46 \pm 0.63}$ | $25.15 \pm 0.89$ | $44.72 \pm 1.39$ | N/A |
| **MassSpecGym (Bushuiev et al., 2024a) results** | | | | | | | |
| Random | $00.37$ | $02.01$ | $08.22$ | $03.06$ | $11.35$ | $27.74$ | - |
| MIST[2] | $\mathbf{14.64}$[2] | $\mathbf{34.87}$[2] | $\mathbf{59.15}$[2] | $09.57$ | $22.11$ | $41.12$ | - |

[1]: Obtained using a different temperature value ($\tau = 4$) than used for retrieval scores ($\tau = 1/256$), as optimal values for the two objectives drastically differ. For more details, see Appendix Table 3.

[2]: MIST scores were obtained by running the model with ground truth molecular formulas. In the mass-based retrieval setting, this can be interpreted as either a form of data leakage, or as an "oracle" setting in which the formula is known. For more information, see MassSpecGym GitHub Issue #52.

dictions are binarized using $\hat{y}^{(i)} > 0.5$. For computing retrieval scores, continuous $[0, 1]$-bounded predictions are used.

**Experimental results.** Table 1 shows test performances of all eight tested loss functions in terms of molecular retrieval and average Tanimoto similarity. In addition, validation performances in function of training time for a selection of loss functions are shown in Appendix C (Figure 3). Best retrieval scores are obtained with contrastive losses. Among those, we empirically find the versions using the cosine similarity score (either on the fingerprint, or on the embedding level) to outperform the learned MLP similarity variants. The best average Tanimoto similarities are obtained using the IoU Loss. These results are somewhat expected, as these two types of losses can be seen as specifically optimizing for these respective metrics. However, we empirically find that the two objectives of retrieval and fingerprint accuracy (in terms of Tanimoto similarity) seem to oppose one another, as the best scoring models in each category are conversely the worst scoring in the other. This result is somewhat surprising: to predict more accurate fingerprints is to worsen retrieval results. This fundamental trade-off presents itself as a pareto front, also visualized in Figure 1 in the introduction, where results from Table 1 are shown as a scatterplot.

The above results were obtained after carefully tuning the considered methods. Tuning results for focal loss $\gamma$ and contrastive temperature $\tau$ are shown in Appendix C (Tables 2 and 3). All results used the optimal hyperparameter values chosen in terms of HR@20. An exception to this rule was made for the contrastive Fp-Cos model, as an inverse correlation was found between retrieval scores and Tanimoto similarities in function of the temperature hyperparameter. For this reason, two versions of models using this loss were trained separately using optimal temperatures for both metrics.

We also tested the impact of the similarity function used during retrieval (i.e., during evaluation/inference after a model is trained). In practice, retrieval results are better when using cosine similarity to score candidates against a prediction, irregardless of the loss function used during training (see Appendix C – Table 4). Hence, the results shown in Table 1 compute retrieval scores at inference time using the cosine similarity score between predicted fingerprint and candidates. An exception to this rule are the contrastive loss functions, where retrieval scores are inferred using the similarity function part of the neural network (i.e., $g$). In addition, we present experimental results using (1) different choices for negative candidates sets in Appendix D, and (2) using weighted combinations of contrastive and IoU loss terms in Appendix C (Figure 4).

## 4 THEORETICAL ANALYSIS OF LOSS FUNCTIONS

The previous empirical results suggest an apparent paradox: "better" retrieval scores can only be obtained at the expense of "worse" similarity scores. To explain how this phenomenon arises, a theoretical investigation of the used loss functions is warranted. Adopting a decision-theoretic perspective (Devroye et al., 2013; Bartlett et al., 2006), we first discuss Bayes-optimal inference for the loss functions discussed previously. In the decision-theoretic (plug-in) approach, one first fits a probabilistic model (often via cross-entropy) and then decodes by maximizing the target metric's expected value. In multi-class and multi-label settings this Bayes-risk view cleanly explains what information about the underlying distribution is needed to be optimal under different metrics, and permits regret analyses when a different metric is optimized instead. We conduct such a regret analysis in the second part of this section.

A different approach is to minimize a non-differentiable task loss during training via surrogates, such as structured SVMs/hinges or other differentiable relaxations (Hazan et al., 2010). The methods based on vectorwise losses in Section 2.2 belong to this category. A long line of work studies when such surrogates are consistent for the target metric and how their regret transfers to metric regret. Since our focus in this paper is rather on comparing models that optimize different losses, such an analysis is out of scope for this paper. Theoretical and empirical studies (e.g. (Ye et al., 2012; Dembczynski et al., 2013)) show that decision-theoretic and direct utility maximization approaches can be asymptotically equivalent, but differ in finite data regimes and under model misspecification.

### 4.1 BAYES-RISK ANALYSIS

To perform Bayes-risk analyses, we assume that training and test data is i.i.d. sampled from an unknown probability distribution $P$ over $\mathcal{X} \times \mathcal{Y}$. The associated ground-truth conditional distribution will be denoted by $p(\boldsymbol{y} \mid \boldsymbol{x})$. Specifically for metabolomics data, one can assume that $p(\boldsymbol{y} \mid \boldsymbol{x}^{(i)}) = 0$ for all $\boldsymbol{y}$ that are not in the candidate set $\mathcal{C}^{(i)}$. For a spectrum $\boldsymbol{x}$ and performance metric $U : \mathcal{Y} \times \mathcal{Y} \to \mathbb{R}$, the prediction $\boldsymbol{y}^\star$ that maximizes the expected value of the metric (i.e., the Bayes-optimal decision) is given by:

$$\boldsymbol{y}_U^\star(\boldsymbol{x}) = \arg\max_{\hat{\boldsymbol{y}} \in \{0,1\}^m} \sum_{\boldsymbol{y} \in \mathcal{C}} U(\hat{\boldsymbol{y}}, \boldsymbol{y}) \, p(\boldsymbol{y} \mid \boldsymbol{x}) \,. \tag{9}$$

In the following, we (1) define the Bayes-optimal decision for each utility metric of interest, and (2) discuss how each loss function can be regarded as a smooth surrogate for the corresponding metric.

**Bitwise loss functions.** Define bitwise accuracy as $\frac{1}{m} \sum_{i=1}^m [y_i = \hat{y}_i]$ with $[\cdot]$ the indicator function. Dembczyński et al. (2012) have shown that the Bayes-optimal decision for bitwise accuracy is:

$$y_i^\star(\boldsymbol{x}) = \arg\max_{b \in \{0,1\}} \mathbb{P}(Y_i = b \mid \boldsymbol{x}) = \arg\max_{b \in \{0,1\}} \sum_{\boldsymbol{y} \in \mathcal{Y}: y_i = b} p(\boldsymbol{y} \mid \boldsymbol{x}) \,,$$

$\mathbb{P}(Y_i = b \mid \boldsymbol{x})$ is the marginal probability of the $i$-th bit being $b$, so binary cross entropy (1) or focal loss (2) can therefore be interpreted as surrogates for bit-wise accuracy (Bartlett et al., 2006).

**Vectorwise loss function.** As performance metrics in multi-label classification tasks, the cosine similarity, Tanimoto similarity and F1-measure are very related[3]. All three lead to fractions with the same numerator, but a different denominator. For F1-measure it has been shown in the past that the Bayes-optimal prediction can be obtained by analyzing $m^2 + 1$ parameters of the joint distribution, but no closed-form expression exists (Waegeman et al., 2014). For Tanimoto similarity, no efficient way of finding the Bayes-optimal prediction exists, and the inference problem is believed to be NP-hard (Chierichetti et al., 2010). A widely adopted training-time remedy is to replace cross-entropy with a differentiable sigmoid-smoothed version of Tanimoto (as done in Section 2.2 and our experiments), or to consider a convex Lovász extension of the IoU loss, yielding the Lovász-Softmax surrogate (Berman et al., 2017).

---

[3]In multi-label classification, those three metrics can be computed per instance (instance-wise averaging), per label (macro-averaging), or by pooling all predictions together for a dataset (micro-averaging). Bayes-optimal decisions might differ for the three versions, but only the intance-wise averaged version is relevant for this paper.

The cosine similarity loss has only become popular in recent years, due to advances in self-supervised learning (Chen et al., 2020). While the empirical success is clear, the decision-theoretic literature on Bayes-optimal prediction under cosine utility is nascent compared to F1-measure and Tanimoto. As shown in a first concrete theoretical result (with formal proof in Appendix E), it turns out that computing the Bayes-optimal decision is simpler than for F1-measure and Tanimoto.

**Theorem 1.** *For $i \in \{1, \ldots, m\}$ define $u_i = \sum_{\boldsymbol{y} \in \{0,1\}^m} \frac{y_i\, p(\boldsymbol{y} \mid \boldsymbol{x})}{\|\boldsymbol{y}\|}$ and let $u_{(1)} \geq \cdots \geq u_{(m)}$ be the sorted weights. For any fixed size $s$, the best $s$-sparse Bayes predictor for cosine similarity selects the $s$ indices with largest $u_i$'s (ties arbitrary). Overall, the optimal support size is $s^{\star} \in \arg\max_{s \in \{1,\ldots,m\}} \frac{1}{\sqrt{s}} \sum_{t=1}^{s} u_{(t)}$.*

**Listwise loss functions.** For HR@1 the Bayes-optimal decision is known to be the mode of the joint distribution (Dembczyński et al., 2012):

$$\boldsymbol{y}^{\star}_{\text{HR@1}}(\boldsymbol{x}) = \arg\max_{\boldsymbol{y} \in \mathcal{C}} p(\boldsymbol{y} \mid \boldsymbol{x}).$$

This is also known as the MAP estimate. In Appendix F we explain how the contrastive loss in (4) is a computationally efficient alternative to categorical cross-entropy, as a surrogate for HR@1 maximization. Furthermore, we also discuss Bayes-optimal decisions for HR@k.

These theoretical results already provide justification for why the different methods analyzed in the experiments cannot be optimal for all metrics at the same time. To illustrate that Bayes-optimal decisions can be different, let us consider a simple example with fingerprint bitvectors of length 5 and $l = 4$ non-zero probability candidates ($\mathcal{C} = \{\boldsymbol{y}_1, \ldots \boldsymbol{y}_4\}$) – see table on the right. For this toy example, it is easy to exhaustively enumerate all possible combinations and obtain the Bayes-optimal solution by brute force. Using the values provided, we find that $\boldsymbol{y}^{\star}_{\text{HR1}} = 11001$, $\boldsymbol{y}^{\star}_{\text{BIT}} = 10000$, $\boldsymbol{y}^{\star}_{\text{COS}} = 11111$,

| $\boldsymbol{y}_i$ | $p(\boldsymbol{y} \mid \boldsymbol{x})$ |
|---|---|
| $\boldsymbol{y}_1 = 11000$ | 0.05 |
| $\boldsymbol{y}_2 = 10100$ | 0.25 |
| $\boldsymbol{y}_3 = 10010$ | 0.30 |
| $\boldsymbol{y}_4 = 11001$ | 0.40 |

and $\boldsymbol{y}^{\star}_{\text{IOU}} = 11001$. This illustrates that, even in such simple cases, the optimal solutions under different metrics can diverge substantially. Our example connects to the notion of "regret", i.e. the difference between the expected loss of the chosen solution and the minimal achievable Bayes loss (Berger, 2013). In the next section, we generalize this example and derive regret bounds.

## 4.2 REGRET ANALYSIS

In this section, we conduct a regret analysis of Bayes-optimal decision rules under different loss functions discussed in Section 4.1 and present novel bounds. First, we define the regret of decoding with rule $V$ but evaluating under metric $U$ as

$$\mathcal{R}^{\text{vs}\, V}_U := \mathbb{E}[U(\boldsymbol{Y}, \boldsymbol{y}^{\star}_U(\boldsymbol{x})) \mid \boldsymbol{x}] - \mathbb{E}[U(\boldsymbol{Y}, \boldsymbol{y}^{\star}_V(\boldsymbol{x})) \mid \boldsymbol{x}],$$

and its worst-case regret over any probability distribution is denoted as $\sup_{p(\cdot \mid \boldsymbol{x})} \mathcal{R}^{\text{vs}\, V}_U$.

All the introduced bounds are conditioned on a fixed spectrum $\boldsymbol{x}$ and candidate set $\mathcal{C} = \{\boldsymbol{y}_1, \ldots, \boldsymbol{y}_\ell\}$ with conditional probabilities $p_j := p(\boldsymbol{y}_j \mid \boldsymbol{x})$ and $\sum_{j=1}^{\ell} p_j = 1$. Let $p_{\star} := \max_j p_j$ denote the highest conditional probability. For a similarity metric $U_{\text{sim}}(\boldsymbol{y}_i, \boldsymbol{y}_j) \in [0, 1]$ such that $U_{\text{sim}}(\boldsymbol{y}_i, \boldsymbol{y}_i) = 1$ and $U_{\text{sim}}(\boldsymbol{y}_i, \boldsymbol{y}_j) < 1$ for $i \neq j$, we also define the similarity band:

$$\sigma_{\min} := \min_{i \neq j} U_{\text{sim}}(\boldsymbol{y}_i, \boldsymbol{y}_j), \qquad \sigma_{\max} := \max_{i \neq j} U_{\text{sim}}(\boldsymbol{y}_i, \boldsymbol{y}_j),$$

A first novel regret bound concerns the regret regarding any vectorwise similarity metric (e.g. Tanimoto, cosine, F1-measure, etc.) when maximizing HR@1 instead.

**Theorem 2** (Regret of HR@1 under vectorwise similarity). *Under the additional assumption that the similarity metric is symmetric, $U_{\text{sim}}(\boldsymbol{y}_i, \boldsymbol{y}_j) = U_{\text{sim}}(\boldsymbol{y}_j, \boldsymbol{y}_i)$, the regret of HR@1 decoding evaluated under the similarity metric $U_{\text{sim}}$ obeys:*

$$\mathcal{R}^{\text{vs}\,\text{HR@1}}_{\text{sim}} \leq [(\sigma_{\max} - \sigma_{\min})(1 - p_{\star} - p_{\text{sim}}) - (1 - \sigma_{\max})(p_{\star} - p_{\text{sim}})]_+ = \sup_{p(\cdot \mid \boldsymbol{x})} \mathcal{R}^{\text{vs}\,\text{HR@1}}_{\text{sim}},$$

*where $p_{\text{sim}} = p(\boldsymbol{y}^{\star}_{\text{sim}}(\boldsymbol{x}) \mid \boldsymbol{x})$ and $[z]_+ := \max\{0, z\}$.*

The above upper bound is tight if no additional assumptions on the candidate set are made. We present the proof of the theorem and its tightness in Appendix G.1. The result shows that regret increases with conditional uncertainty $(1 - p_\star)$ and with the width of the candidate-set similarity band $(\sigma_{\max} - \sigma_{\min})$. A larger margin between $p_\star$ and $p_{\text{sim}}$ reduces the regret, but its influence is attenuated when $\sigma_{\max}$ is close to 1 (near duplicates in the candidate set), because the subtractive term involves $(1 - \sigma_{\max})$. Empirically, in the equal-mass candidate sets of MassSpecGym, we often observe a large range of bands and sometimes $\sigma_{\max} \approx 1$, as shown in Appendix C (Figures 5 and 6). This makes the HR@1 choice particularly vulnerable under Tanimoto/cosine similarity, which is consistent with the Pareto trade-off in Figure 1 and Table 1.

A second novel regret bound concerns the regret regarding HR@1 when maximizing any similarity metric (or minimizing any vectorwise loss) instead.

**Theorem 3** (Regret of vectorwise similarity under HR@1)**.**

$$\mathcal{R}_{\text{HR@1}}^{\text{vs sim}} \leq \min \left\{ p_\star, \frac{\sigma_{\max} - \sigma_{\min}}{1 - \sigma_{\max}} (1 - p_\star) \right\} .$$

We present the proof of the theorem in Appendix G.2. Again, wide similarity bands and near duplicates ($\sigma_{\max} \to 1$) inflate this regret. In data regimes with high uncertainty (small $p_\star$), the second term becomes dominant and can be large, explaining why maximizing Tanimoto/cosine similarity may substantially hurt HR@1—as observed for IoU-trained models in Table 1. Based on this results we can also derive a sufficient condition for no regret (agreement of decisions) between two rules – see Appendix G.3 for details.

The last case we consider in regret analysis is the gap between decoding for bitwise accuracy evaluated under HR@1 and vectorwise loss functions. While the bitwise accuracy can also be viewed as a vectorwise loss, and thus its relation to HR@1 regret is covered by Theorem 3, an alternative general worst case result is known (Dembczyński et al., 2012): $\sup_{p(\cdot|\boldsymbol{x})} \mathcal{R}_{\text{HR@1}}^{\text{vs BIT}} = \frac{1}{2}$. Following this line of constructing regret bounds, we also show that decoding by bitwise accuracy (optimized by BCE/focal) can be arbitrarily misaligned with other vectorwise losses of interest:

**Theorem 4** (Regret bitwise-loss under vectorwise similarity)**.**

$$\sup_{p(\cdot|\boldsymbol{x})} \mathcal{R}_{\text{Tan}}^{\text{vs Bit}} \geq \tfrac{1}{2}, \qquad \sup_{p(\cdot|\boldsymbol{x})} \mathcal{R}_{\text{Cos}}^{\text{vs Bit}} \geq \tfrac{1}{2} \quad (m \geq 3) .$$

Appendix G.4 contains the proofs of these supremas, as well as alternative bounds on $\sup \mathcal{R}_{\text{Bit}}^{\text{vs HR@1}}$, $\sup \mathcal{R}_{\text{Cos}}^{\text{vs HR@1}}$ and $\sup \mathcal{R}_{\text{Tan}}^{\text{vs HR@1}}$ (depending only on the number of fingerprint bits $m$). In Appendix I, we discuss how all these regret analyses can be expanded to include finite-sample generalization error components.

### 4.3 PRACTICAL IMPLICATIONS OF OUR THEORETICAL ANALYSIS

All theoretical results presented in this section are entirely novel, and hold implications for the future development of computational metabolomics methods. Specifically, while our empirical results in Section 3 use Morgan ($r = 2$, 4096-bit) fingerprints, the regret analyses in Section 4.2 abstracts away from specific fingerprint choice. Instead, they rely on a notion of "similarity bands", meaning that different binary molecular representations naturally fit into the same framework. As such, our theoretical results may be used to inform fingerprint selection. Experiments with five alternative fingerprints (Morgan ($r \in 4, 6, 8$), MAP4 (Capecchi et al., 2020), RDKit (Landrum, 2013)) are reported in Appendix H; we summarize the outcomes here.

To motivate alternative fingerprint selection, consider that minimizing regret (and, hence, negating the trade-off) is favorable. After all, many methods that predict fingerprints still primarily optimize for fingerprint similarity (Baygi & Barupal, 2024). Lower regret implies that improving prediction similarity more directly boosts retrieval performance. Our theory shows that the regret when maximizing vectorwise similarity w.r.t. HR@1 (and vice versa) depend on both (1) conditional probability distributions $p(\boldsymbol{y} \mid \boldsymbol{x})$, and (2) $\sigma_{\min}$ and $\sigma_{\max}$. While the former are unknown, the latter can be computed from data (as in Appendix C, Figure 5), making them potentially useful for model design.

To illustrate, we trained models with IoU and Contrastive (Emb-Cos) losses across all alternative fingerprints. The results align with theory: fingerprints for which the theory suggests that regret should be lower, show smaller differences in HR@1 between both types of loss functions (Appendix H, Table 7). Among those, MAP4 performs competitively or better than Morgan ($r = 2$), suggesting it is a promising complementary or alternative fingerprint prediction target for computational metabolomics.

## 5  DISCUSSION

In this study we observed a consistent Pareto trade-off between (1) fingerprint similarity and (2) molecular retrieval: improving one typically degrades the other (Figure 1). Using Bayes-risk analysis, we have theoretically explained why these outcomes are expected, and defined regret bounds on how different the optimal decisions by each model may be. In this

Most concurrent research uses modeling strategies that optimize for predicting correct fingerprints (Dührkop et al., 2015; Goldman et al., 2023). All the while, these models have molecular retrieval as their implicit goal. Our experimental results straightforwardly show that—if the goal is to perform molecular retrieval—the loss function should be chosen with this task in mind (i.e., using a contrastive loss). Moreover, our theoretical results confirm that the regret between the decisions for either (1) optimal similarity, or (2) optimal retrieval, can be large. It is trivial to show that, when the conditional class distribution is degenerate (i.e., $p(\boldsymbol{y} \mid \boldsymbol{x}) = 1$ for the true candidate), there is no regret. As such, our empirical results can only be explained in the context of assuming uncertainty in the output class distribution. It is known that molecular retrieval is a generally hard problem because of both (1) the insufficiency of signal in the input mass spectra, and (2) the vast complexity of chemical space (da Silva et al., 2015). These two issues connect to the notions of aleatoric and epistemic uncertainty, respectively (Hüllermeier & Waegeman, 2021). For this reason, we consider uncertainty estimation to be a valuable future research direction in this field.

We derived regret bounds that generalize for any similarity function and even choice of fingerprint bitvector representation. As such, we are convinced that the trade-off is not a product of our specific experimental setup. Moreover, Theorems 2 and 3 show that, for some choices of similarity function and/or fingerprints type, the regret may actually be small. Minimizing regret is desirable, as obtaining models excelling at both tasks is practically useful. This connects to an emerging are of research broadly investigating how the choice of molecular representation impacts model performance (Huber & Pollmann, 2025). In Appendix H, we presented results using alternative fingerprints, revealing that MAP4 performs comparably to Morgan ($r = 2$) fingerprints, but with lower regrets. We anticipate that these theoretical insights will inform subsequent research on fingerprint selection.

### REPRODUCIBILITY STATEMENT

The following significant efforts have been undertaken to ensure the reproducibility of this manuscript. Methods are described in the main text, as well as elaborated fully upon in the Appendices. Complete proofs for all theorems in the main text are also present in the Appendices. Source code to reproduce all experimental results are co-submitted as a supplementary zip file. In addition, the source code is structured as a PyPI package, which will be published as a pip-installable package for easy use upon publication. Together, these measures should fully ensure the reproducibility of our work.

### LLM USAGE

This manuscript made use of LLMs as a general-purpose assist tool. The precise ways that LLMs were used are the following: (1) correcting the flow of writing, (2) scanning mathematical proofs written by hand on paper and converting them to initial draft LaTeX code, and (3) searching for relevant literature.

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

## A  SUPPLEMENTARY METHODS

**Model architecture.**  The full fingerprint prediction model, denoted by $f$, produces predictions $\hat{\boldsymbol{y}} \in [0,1]^m$ according to:

$$\hat{\boldsymbol{y}}^{(i)} = f(\boldsymbol{x}^{(i)}) = \sigma(\boldsymbol{W}_{\text{out}} E(\boldsymbol{x}^{(i)})),$$

where $\sigma$ denotes the sigmoid operation, and $\boldsymbol{W}_{\text{out}} \in \mathbb{R}^{m \times d}$ denotes the learnable weights of the last linear layer. All other neural network layers are included in $E$. In all experiments, $E$ uses the same MLP architecture consisting of 3 fully-connected layers. Inbetween each layer, a GeLU activation Hendrycks & Gimpel (2016), dropout Srivastava et al. (2014) (using a rate of 0.25) and Layer-Norm operation Ba et al. (2016) is placed, in that order. The first layer maps the 10050-dimensional space to 1024 dimensions, which is also the hidden dimensionality used in all subsequent layers. More formally, let $B(\cdot) = \text{LayerNorm}(\text{Dropout}(\text{GeLU}(\cdot), 0.25))$. the neural network embedder $E$ is, then, described fully as:

$$E(\boldsymbol{x}^{(i)}) = \boldsymbol{W}_3 \cdot B(\boldsymbol{W}_2 \cdot B(\boldsymbol{W}_1 \cdot B(\boldsymbol{W}_1 \boldsymbol{x}^{(i)})))$$

with $\boldsymbol{W}_1 \in \mathbb{R}^{1024 \times 10050}$, $\boldsymbol{W}_2 \in \mathbb{R}^{1024 \times 1024}$, and $\boldsymbol{W}_3 \in \mathbb{R}^{1024 \times 1024}$, the learnable weight matrices of the linear layers (with biases omitted for simplicity).

Models were trained using one of the eight loss functions explained in Section 2.2: (1) BCE, (2) FL, (3) Cosine Loss, (4) IoU Loss, (5) Contrastive FP-Cos, (6) Contrastive Emb-Cos, (7) Contrastive Fp-MLP, and (8) Contrastive Emb-MLP. For the contrastive MLP variants, additional layers are also fitted that either take (1) the predicted and true fingerprint as input ("Fp-MLP"), or (2) an embedding of the spectrum and candidate molecule as input ("Emb-MLP"). In the former case, the MLP is of the form:

$$\text{MLP}(\hat{\boldsymbol{y}}, \boldsymbol{c}_j) = \boldsymbol{W}_3 \cdot B(\boldsymbol{W}_2 \cdot B(\boldsymbol{W}_1(\hat{\boldsymbol{y}} \,\|\, \boldsymbol{c}_j \,\|\, \hat{\boldsymbol{y}} \odot \boldsymbol{c}_j))).$$

The input to this similarity-learning MLP, hence, uses a concatenation of (1) the predicted fingerprint $\hat{\boldsymbol{y}} \in [0,1]^{4096}$, (2) a candidate fingerprint $\boldsymbol{c}_j \in \{0,1\}^{4096}$, and (3) their element-wise product $\hat{\boldsymbol{y}} \odot \boldsymbol{c}_j \in [0,1]^{4096}$. The MLP uses learnable linear layers with weight matrices (biases omitted for simplicity): $\boldsymbol{W}_1 \in \mathbb{R}^{1536 \times 12288}$, $\boldsymbol{W}_2 \in \mathbb{R}^{768 \times 1536}$, and $\boldsymbol{W}_3 \in \mathbb{R}^{1 \times 768}$. The function $B$ denotes the same block as in the main fingerprint-producing MLP: $B(\cdot) = \text{LayerNorm}(\text{Dropout}(\text{GeLU}(\cdot), 0.25))$. In the latter case ("Emb-MLP"), the MLP is of a similar form. Let us denote the spectrum embedding and candidate embedding as $\boldsymbol{e}_{\text{spec}} = \boldsymbol{W}_{\text{spec}} E(\boldsymbol{x})$ and $\boldsymbol{e}_{\text{cand},j} = \boldsymbol{W}_{\text{cand}} \boldsymbol{c}_j$. The MLP is the "Emb-MLP" contrastive model is, then:

$$\text{MLP}(\boldsymbol{e}_{\text{spec}}, \boldsymbol{e}_{\text{cand},j}) = \boldsymbol{W}_3 \cdot B(\boldsymbol{W}_2 \cdot B(\boldsymbol{W}_1(\boldsymbol{e}_{\text{spec}} \,\|\, \boldsymbol{e}_{\text{cand},j} \,\|\, \boldsymbol{e}_{\text{spec}} \odot \boldsymbol{e}_{\text{cand},j}))),$$

where $\boldsymbol{W}_1 \in \mathbb{R}^{96 \times 768}$, $\boldsymbol{W}_2 \in \mathbb{R}^{48 \times 96}$, and $\boldsymbol{W}_3 \in \mathbb{R}^{1 \times 48}$ denote the learnable linear weight matrices (biases omitted for simplicity).

**Main experiments.**  All models were trained with the Adam optimizer in float-32 precision using a batch size of 64 for a maximum of 50 epochs. Gradients were clipped to a norm of 1. A variable learning rate (however, fixed during one run) was used depending on the hyperparameter configuration (see below). After every 1000 training steps, validation performance was checked in terms of: (1) HR@1, (2) HR@5, (3) HR@20, and (4) Tanimoto similarity. For each of these metrics, the checkpoint was saved for when that metric was optimal (i.e., maximal). For every model, the appropriate checkpoint was used to compute final validation and test performance. For example, reported validation and test HR@20 scores were always derived from model checkpoints with optimal validation HR@20s during their training runs. This was performed because the convergence in terms of Tanimoto similarity may not coincide with the convergence in terms of retrieval. For bitwise and vectorwise loss functions, hit rates were computed (and checkpoints made) for both retrieval settings. For contrastive loss functions—as these models use the candidates during training—separate model runs were performed for the two retrieval settings.

For every loss function, five different learning were tested: $\{0.00005, 0.00007, 0.0001, 0.0003, 0.0005\}$. For each of these possible learning rates, 5 models were trained. The best learning rate was then chosen based on the highest average HR@20 that learning rate achieved. All other metrics are then reported using the checkpoints derived from those 5 models. For the focal loss, the $\gamma$ hyperparameter was additionally tuned in the same manner, but using a fixed learning rate of 0.0007 (a value chosen because it was the optimal value for the BCE loss). Results are in Appendix Table 2.

For the contrastive loss function, the same was performed for different values of $\tau$, results are in Appendix Table 3. Upon finding the optimal value of their respective hyperparameters, an identical procedure was performed for finding the optimal learning rate and reporting results as with the other models.

# B  RELATED WORK

Early computational approaches for small-molecule identification from MS/MS data framed the task as molecular property prediction, in which a model predicts a binary structural fingerprint used for database retrieval. A foundational line of work is CSI:FingerID (Dührkop et al., 2015), which trains a large collection of support vector machines to predict individual fingerprint bits using molecular fragmentation tree-based inputs. CSI:FingerID is optimized explicitly for bitwise accuracy, and its fingerprints are used with Tanimoto similarity for retrieval. Although influential, CSI:FingerID predates large supervised datasets such as MassSpecGym, and its learning setup (bit-independent classifiers, handcrafted features, and no end-to-end neural representation) differs substantially from the deep learning architectures evaluated in this study. Importantly, CSI:FingerID optimizes each bit in isolation, placing it firmly in the "bitwise losses" category examined here.

More recent work has moved toward neural models that learn joint representations of spectra and structures. MIST (Goldman et al., 2023) represents the most relevant contemporary baseline. MIST trains a chemical-formula-aware transformer to predict fingerprints in two stages. First, it optimizes for fingerprint accuracy using the Cosine loss, after which a fine-tuning stage optimizes for retrieval with a contrastive loss. Final retrieval is performed via embedding-space cosine similarity rather than via predicted binary fingerprints. Because MIST is trained with listwise contrastive objectives and uses learned projections of candidate fingerprints, its final optimization target aligns directly with HR@1 retrieval, not with fingerprint similarity. Our contrastive Emb-Cos variants mirror the contrastive fine-tuning setup of MIST. Interestingly, although not elaborated there, MIST observes a similar trade-off to the one studied in this manuscript. Specifically, the initial fingerprint predictor optimized with Cosine loss outperforms CSI:FingerID in fingerprint prediction accuracy but underperforms it in retrieval. For the purpose of examining this trade-off in better detail, we have chosen a simpler architectural set-up. Specifically, our work uses a simple MLP over binned spectra rather than a chemical formula transformer. Table 1 shows that well-tuned variants of such MLPs can still obtain competitive (even state-of-the-art) performances.

Beyond these two lines of work, other recent methods (e.g., CMSSP, Chen et al. (2024), JESTR, Kalia et al. (2025)) also employ contrastive or metric-learning formulations, again emphasizing retrieval-optimized training objectives over fingerprint accuracy. Collectively, this emerging trend aligns with our empirical and theoretical results: methods directly optimizing a retrieval-aligned objective (contrastive/listwise losses) outperform fingerprint-centric objectives, but often at the expense of fingerprint similarity. Our work is the first to systematically quantify and theoretically explain this trade-off, providing a framework that clarifies when and why approaches such as MIST, CSI:FingerID, and contrastive neural retrieval models differ in behavior on benchmarks like MassSpecGym.

## C FIGURES AND TABLES SUPPORTING THE RESULTS SECTION

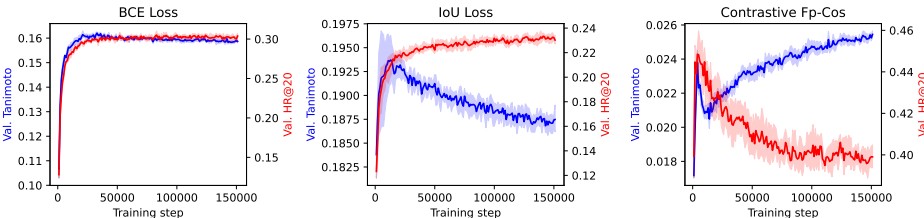

Figure 3: Validation performance (in terms of Average Tanimoto, blue, and HR@20, red) over training time. Curves are shown for the BCE Loss, IoU Loss, and Contrastive FP-Cos, respectively. Note that the Contrastive FP-Cos curve is obtained using a temperature of 1/256. As such, the model is tuned for better hit rates (see Table Table 3). In addition, note that our experimental setup accounts for different optima in time w.r.t. different metrics, as final scores are reported using the checkpoint for which each metric was optimal.

Table 2: Focal loss $\gamma$ tuning results (Bushuiev et al., 2024a). All performances are validation scores. Numbers report averages and standard deviations over 5 model runs. All 20 models used in this table were trained with a learning rate of 7e-5. Retrieval scores are those using equal mass candidates. The best hyperparameter value (bold) was chosen based on retrieval scores.

| Focal $\gamma$ | Tanimoto (IoU) | HR@1 | HR@5 | HR@20 |
|---|---|---|---|---|
| 1.0 | $\mathbf{16.27 \pm 0.07}$ | $08.94 \pm 0.18$ | $18.14 \pm 0.37$ | $32.45 \pm 0.33$ |
| **2.0** | $16.20 \pm 0.06$ | $\mathbf{09.20 \pm 0.08}$ | $\mathbf{18.74 \pm 0.20}$ | $\mathbf{33.05 \pm 0.16}$ |
| 5.0 | $15.88 \pm 0.10$ | $08.45 \pm 0.18$ | $18.10 \pm 0.25$ | $31.53 \pm 0.32$ |
| 10.0 | $15.70 \pm 0.05$ | $06.31 \pm 0.23$ | $15.00 \pm 0.33$ | $27.72 \pm 0.27$ |

Table 3: Contrastive loss temperature $\tau$ tuning results (Bushuiev et al., 2024a). All performances are validation scores. Numbers report averages and standard deviations over 5 model runs. All models used in this table were trained with a learning rate of 7e-5. Tanimoto scores are only reported for the "FP-Cos" contrastive model, as this is the only one that can be interpreted as predicting a true fingerprint. Retrieval scores are those using equal mass candidates. Best hyperparameter values (bold) were chosen based on retrieval scores.

| Contrastive loss type | Temperature $\tau$ | Tanimoto (IoU) | HR@1 | HR@5 | HR@20 |
|---|---|---|---|---|---|
| FP-Cos | **1/256** | $02.67 \pm 0.03$ | $\mathbf{11.31 \pm 0.59}$ | $\mathbf{25.51 \pm 1.02}$ | $\mathbf{45.60 \pm 1.09}$ |
| | 1/64 | $04.06 \pm 0.26$ | $11.17 \pm 0.18$ | $24.75 \pm 0.62$ | $44.54 \pm 0.65$ |
| | 1/16 | $03.63 \pm 0.05$ | $09.75 \pm 0.29$ | $21.42 \pm 0.25$ | $37.66 \pm 0.29$ |
| | 1/4 | $06.66 \pm 0.05$ | $08.23 \pm 0.18$ | $18.32 \pm 0.26$ | $34.13 \pm 0.23$ |
| | 1 | $07.19 \pm 0.04$ | $08.17 \pm 0.12$ | $18.06 \pm 0.18$ | $33.48 \pm 0.15$ |
| | 4 | $\mathbf{07.38 \pm 0.04}$ | $08.14 \pm 0.10$ | $18.20 \pm 0.19$ | $33.77 \pm 0.16$ |
| Emb-Cos | 1/256 | N/A | $12.04 \pm 0.65$ | $26.43 \pm 0.62$ | $45.43 \pm 0.62$ |
| | **1/64** | N/A | $\mathbf{12.08 \pm 0.28}$ | $\mathbf{27.42 \pm 0.75}$ | $\mathbf{47.44 \pm 0.78}$ |
| | 1/16 | N/A | $10.76 \pm 0.15$ | $24.29 \pm 0.23$ | $45.68 \pm 0.51$ |
| | 1/4 | N/A | $06.18 \pm 0.16$ | $17.64 \pm 0.39$ | $39.28 \pm 0.54$ |
| | 1 | N/A | $04.08 \pm 0.09$ | $14.23 \pm 0.27$ | $33.81 \pm 0.52$ |
| | 4 | N/A | $04.62 \pm 0.32$ | $14.53 \pm 0.41$ | $36.58 \pm 0.86$ |
| Fp-MLP | 1/256 | N/A | $10.66 \pm 0.58$ | $25.13 \pm 0.51$ | $45.85 \pm 0.42$ |
| | 1/64 | N/A | $10.81 \pm 0.58$ | $25.18 \pm 0.84$ | $46.28 \pm 1.08$ |
| | **1/16** | N/A | $\mathbf{12.60 \pm 0.75}$ | $\mathbf{27.84 \pm 0.51}$ | $\mathbf{47.88 \pm 0.70}$ |
| | 1/4 | N/A | $11.53 \pm 0.57$ | $26.93 \pm 0.51$ | $47.21 \pm 0.36$ |
| | 1 | N/A | $12.51 \pm 0.39$ | $27.48 \pm 0.60$ | $47.91 \pm 0.95$ |
| | 4 | N/A | $12.75 \pm 0.19$ | $27.85 \pm 0.68$ | $47.36 \pm 0.65$ |
| Emb-MLP | 1/256 | N/A | $10.75 \pm 0.44$ | $25.05 \pm 0.62$ | $45.89 \pm 0.76$ |
| | 1/64 | N/A | $10.80 \pm 0.48$ | $25.66 \pm 0.89$ | $46.70 \pm 1.08$ |
| | **1/16** | N/A | $\mathbf{11.50 \pm 1.29}$ | $\mathbf{26.33 \pm 1.67}$ | $\mathbf{46.81 \pm 1.31}$ |
| | 1/4 | N/A | $10.85 \pm 0.77$ | $26.01 \pm 1.18$ | $46.71 \pm 0.62$ |
| | 1 | N/A | $11.35 \pm 1.26$ | $25.86 \pm 1.77$ | $46.46 \pm 1.74$ |
| | 4 | N/A | $11.24 \pm 1.61$ | $26.12 \pm 1.89$ | $46.05 \pm 1.54$ |

Table 4: Validation retrieval scores of models using vectorwise loss functions. Only results are shown on the equal mass candidate retrieval setting. On average, performing retrieval using cosine similarity performs better than using Tanimoto similarity (i.e., IoU). Here, IoU retrieval is performed with continuous (i.e., non-binarized) predictions. This holds true even for the model trained using the IoU loss. Scores indicate the average and standard deviation over 5 model runs.

| Model trained using | Evaluate retrieval using | HR@1 | HR@5 | HR@20 |
|---|---|---|---|---|
| Cosine Loss | Cosine | $7.49 \pm 0.16$ | $15.82 \pm 0.21$ | $29.39 \pm 0.21$ |
| | Tanimoto (IoU) | $7.21 \pm 0.21$ | $15.34 \pm 0.14$ | $28.93 \pm 0.25$ |
| IoU Loss | Cosine | $4.29 \pm 0.21$ | $11.00 \pm 0.18$ | $23.66 \pm 0.25$ |
| | Tanimoto (IoU) | $3.86 \pm 0.21$ | $10.45 \pm 0.20$ | $23.01 \pm 0.28$ |

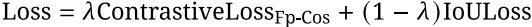

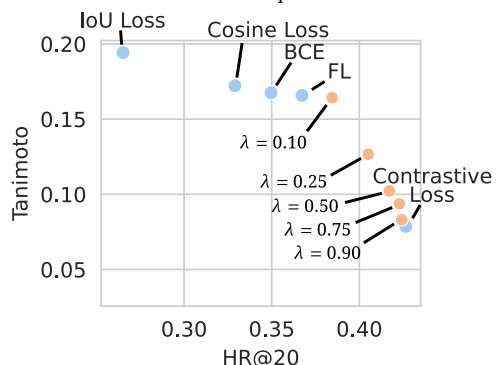

Figure 4: Experimental results when training models using a weighted combination of loss functions. In blue, results are repeated from Figure 1. In orange, results are shown using a combination of contrastive loss and IoU loss, with 5 different weights $\lambda$. Scores indicate the mean over 5 model runs. Notably, Even small weights for on contrastive loss components result in a marked increase in retrieval scores over non-contrastive loss functions.

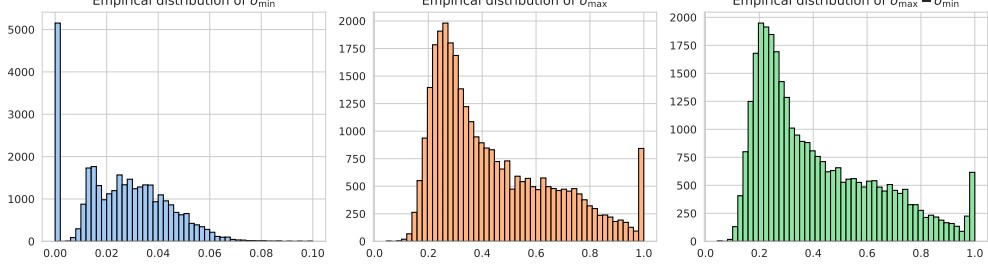

Figure 5: Empirical distribution of $\sigma_{\min}$, $\sigma_{\max}$, and $\sigma_{\max} - \sigma_{\min}$ in all MassSpecGym equal-mass candidate sets using the **Tanimoto similarity** on Morgan Fingerprints (4096 bits, radius 2) to compare candidates to true molecules. $\sigma_{\min}$ denotes the lowest similarity to the true molecule found in the candidate set. $\sigma_{\max}$ denotes the highest similarity to the true molecule found in the candidate set. In some cases, there are candidates with zero similarity to the true molecule (no fingerprint bits in common). Conversely, in some cases, there exist candidates with an equal fingerprint vector representation (i.e., $\sigma_{\max} = 1$ in some cases). This arises due to the problem of duplicate fingerprints. The median $\sigma_{\min}$ equals 0.026, while the median $\sigma_{\max}$ equals 0.367.

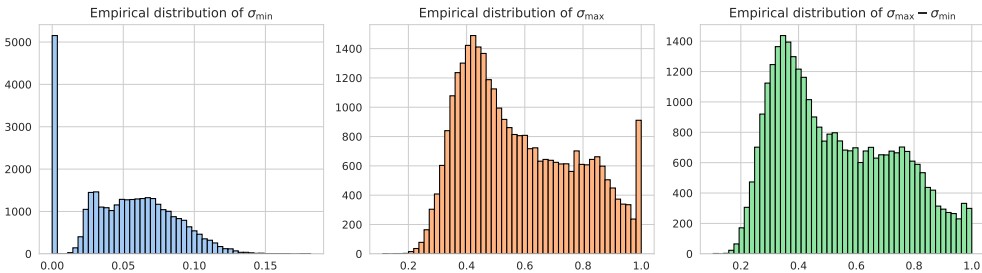

Figure 6: Empirical distribution of $\sigma_{\min}$, $\sigma_{\max}$, and $\sigma_{\max} - \sigma_{\min}$ in all MassSpecGym equal-mass candidate sets using the **Cosine similarity** on Morgan Fingerprints (4096 bits, radius 2) to compare candidates to true molecules. $\sigma_{\min}$ denotes the lowest similarity to the true molecule found in the candidate set. $\sigma_{\max}$ denotes the highest similarity to the true molecule found in the candidate set. In some cases, there are candidates with zero similarity to the true molecule (no fingerprint bits in common). Conversely, in some cases, there exist candidates with an equal fingerprint vector representation (i.e., $\sigma_{\max} = 1$ in some cases). This arises due to the problem of duplicate fingerprints. The median $\sigma_{\min}$ equals $0.053$, while the median $\sigma_{\max}$ equals $0.538$.

## D   THE EFFECT OF CONTRASTIVE NEGATIVE CANDIDATE SET SELECTION

In the main results, contrastive loss function models are trained with the candidate sets provided by MassSpecGym (Bushuiev et al., 2024a). MassSpecGym provides different candidate sets for the equal mass and equal formula candidate scenarios, both capped to a maximum of 256 candidates per input spectrum. In our study, we evaluate each setting by training contrastive models using their respective candidate sets. In this section, we briefly explore how training on different candidate sets influences performance results. All experiments presented in this section are performed using the "Emb-Cos" contrastive model, as this is the one performing best overall.

As a first example, let us investigate how models trained using one candidate set perform when evaluated on the other type of candidate set. In other words, this experiment investigates if a model trained on one candidate set can generalize its retrieval towards other retrieval settings. Here, we take the models trained on equal mass candidates and evaluate their retrieval on equal formula candidates, and vice-versa From Table 5, it can be seen that models trained on equal mass candidates performs better on equal formula candidates than the models trained on equal formula candidates themselves. This can potentially be an effect of the average candidate set being bigger using equal mass candidates. As a consequence, it is possible that equal mass candidate sets contain more "hard" or "representative" negative examples through which a more discriminative model can be trained.

Table 5:   Molecular retrieval and fingerprint accuracy performance on MassSpecGym (Bushuiev et al., 2024a).   The best and second-best performing models for each metric are indicated with boldface and underlined, respectively. For the models trained in this study, numbers report averages and standard deviations over 5 model runs.

| Candidate set origin | Max. (aver.) # cands | Retrieval w. equal mass candidates | | | Retrieval w. equal formula candidates | | |
|---|---|---|---|---|---|---|---|
| | | HR@1 | HR@5 | HR@20 | HR@1 | HR@5 | HR@20 |
| MassSpecGym equal mass | 256 (252.8) | **12.29 ± 0.69** | **26.09 ± 1.43** | **45.05 ± 0.71** | **14.17 ± 0.39** | **28.36 ± 0.46** | **48.86 ± 0.94** |
| MassSpecGym equal formula | 256 (212.1) | 10.43 ± 0.66 | 21.91 ± 1.07 | 39.08 ± 0.72 | 13.62 ± 1.12 | 26.33 ± 1.06 | 46.50 ± 0.94 |
| "Hard" equal mass | 64 (63.9) | 07.74 ± 0.61 | 18.81 ± 0.95 | 36.4 ± 1.62 | 11.23 ± 0.48 | 24.23 ± 0.94 | 44.13 ± 1.26 |
| "Hard" equal mass | 128 (127.7) | 08.13 ± 0.57 | 20.64 ± 1.20 | 39.81 ± 1.70 | 12.25 ± 0.91 | 25.96 ± 0.33 | 45.73 ± 0.62 |
| "Hard" equal mass | 256 (253.8) | 08.34 ± 0.60 | 21.51 ± 1.64 | 40.67 ± 1.58 | 12.30 ± 0.69 | 25.86 ± 1.40 | 44.90 ± 1.24 |
| "Hard" equal mass | 512 (498.2) | 10.81 ± 1.05 | 23.37 ± 0.96 | 41.79 ± 1.33 | 12.38 ± 0.68 | 26.90 ± 1.02 | 45.88 ± 1.17 |
| "Hard" equal mass | 1024 (959.5) | 11.64 ± 0.57 | 23.99 ± 1.38 | 41.89 ± 2.23 | 14.10 ± 0.60 | 27.55 ± 0.85 | 47.15 ± 0.87 |
| "Hard" equal formula | 64 (60.2) | 06.64 ± 0.58 | 17.06 ± 1.76 | 34.69 ± 0.90 | 11.47 ± 0.36 | 24.66 ± 0.36 | 43.60 ± 0.24 |
| "Hard" equal formula | 128 (115.1) | 08.03 ± 0.47 | 18.49 ± 0.95 | 34.82 ± 1.08 | 12.08 ± 0.89 | 25.24 ± 0.57 | 45.04 ± 1.22 |
| "Hard" equal formula | 256 (214.3) | 07.66 ± 0.38 | 19.34 ± 0.80 | 37.74 ± 1.36 | 12.99 ± 0.85 | 26.46 ± 0.59 | 45.49 ± 1.02 |
| "Hard" equal formula | 512 (385.8) | 07.71 ± 0.51 | 19.95 ± 1.73 | 37.52 ± 1.60 | 12.39 ± 0.97 | 25.64 ± 0.90 | 45.38 ± 0.34 |
| "Hard" equal formula | 1024 (667.9) | 08.68 ± 1.59 | 20.80 ± 1.70 | 38.91 ± 1.12 | 12.55 ± 0.46 | 27.08 ± 0.95 | 46.33 ± 1.59 |

To further experiment with the notion of creating informative contrastive candidate sets, we create our own negative candidate sets for training models. Note that, in this case, models are only trained

using these custom negative compound sets, and are still evaluated on the MassSpecGym candidates in the end. For this purpose, for every unique compound in MassSpecGym, we enumerate all compounds with (1) equal formula and (2) equal mass as retrieved in the PubChem 118M compound set included with MassSpecGym. For every retrieved set, we subsequently sort the candidates by decreasing Tanimoto similarity (based on 4096 bit Morgan fingerprints), and only keep the $n$ first candidates. In this way, candidate selection is performed to include the negative compounds which are "hardest" to distinguish by the contrastive model. All models in this experiment were identically trained to those in the main experiments. To only small addition is that for every max number of candidates $n \in \{64, 128, 256, 512, 1024\}$, the temperature was additionally re-tuned (identically as before in the main experiments) between values $\tau \in \{1/256, 1/64, 1/16\}$. We perform this step as varying the number of candidates in the softmax during training may have an impact on optimal temperature values. From Table 5, it can be seen that there is an upward trend in performance when training models with an increasing number of negative candidates per spectrum. Still, performance of models using hard negatives mined from PubChem do not exceed performance in comparison with the MassSpecGym equal mass negative sets.

The results in Table 5 highlight subtle differences in retrieval performance depending on the choice of negative candidate sets. MassSpecGym constructs its negatives in a cascading manner: candidates are first drawn from a set of 1 million biological and environmental molecules, then from a set of 4 million chemically diverse molecules, and only finally from the 118 million molecules in PubChem. Because candidate sets are truncated at a maximum of 256 molecules, in practice most negatives come from the first two sets. In contrast, hard negatives mined from PubChem cover a much broader chemical space, the majority of which are not natural products (i.e., are not biosynthesized) (Kretschmer et al., 2023). In this sense, contrastive models trained on MassSpecGym candidates might only perform well for retrieval within the space of metabolites and environmental molecules (e.g., metabolomics or exposomics). For more open-world annotation tasks, such as the identification of forensic unknowns or patent compounds, it is more realistic to train on PubChem-derived negatives rather than preferentially including metabolites. This perspective recontextualizes the results in Table 5: models trained on 1024 hard PubChem negatives achieve performances comparable to models trained on MassSpecGym negatives in some cases. This trade-off may be justified for applications requiring broader chemical generalization.

## E    PROOF OF BAYES-OPTIMAL PREDICTION FOR COSINE SIMILARITY

We assume $p(\boldsymbol{y} \mid \boldsymbol{x}) = 0$ whenever $\|\boldsymbol{y}\| = 0$, so that $U_{\mathrm{Cos}}(\hat{\boldsymbol{y}}, \boldsymbol{y})$ and the quantities below are well-defined.

**Theorem 1.** *For $i \in \{1, \ldots, m\}$ define $u_i = \sum_{\boldsymbol{y} \in \{0,1\}^m} \frac{y_i \, p(\boldsymbol{y} \mid \boldsymbol{x})}{\|\boldsymbol{y}\|}$ and let $u_{(1)} \geq \cdots \geq u_{(m)}$ be the sorted weights. For any fixed size $s$, the best $s$-sparse Bayes predictor for cosine similarity selects the $s$ indices with largest $u_i$'s (ties arbitrary). Overall, the optimal support size is $s^\star \in \arg\max_{s \in \{1, \ldots, m\}} \frac{1}{\sqrt{s}} \sum_{t=1}^{s} u_{(t)}$.*

*Proof.* For any nonzero $\hat{\boldsymbol{y}} \in \{0,1\}^m$ with support $S(\hat{\boldsymbol{y}}) = \{i : \hat{y}_i = 1\}$ and size $s(\hat{\boldsymbol{y}}) = |S(\hat{\boldsymbol{y}})|$, the expected cosine similarity satisfies

$$\mathbb{E}[U_{\mathrm{Cos}}(\hat{\boldsymbol{y}}, \boldsymbol{y})] = \sum_{\boldsymbol{y}} \frac{\sum_{i=1}^{m} \hat{y}_i y_i}{\|\hat{\boldsymbol{y}}\| \|\boldsymbol{y}\|} \, p(\boldsymbol{y} \mid \boldsymbol{x}) = \frac{1}{\|\hat{\boldsymbol{y}}\|} \sum_{i=1}^{m} \hat{y}_i \sum_{\boldsymbol{y}} \frac{y_i \, p(\boldsymbol{y} \mid \boldsymbol{x})}{\|\boldsymbol{y}\|}$$

$$= \frac{1}{\|\hat{\boldsymbol{y}}\|} \sum_{i=1}^{m} \hat{y}_i u_i = \frac{1}{\sqrt{\sum_i \hat{y}_i}} \sum_{i : \hat{y}_i = 1} u_i,$$

by definition of cosine similarity and linearity of expectation. As a result, maximizing the expected cosine similarity over binary predictions is equivalent to maximizing

$$F(S) = \frac{1}{\sqrt{|S|}} \sum_{i \in S} u_i \qquad \text{over } S \subseteq \{1, \ldots, m\}, \, S \neq \varnothing,$$

where $S$ is the support of the prediction. Let $S^\star$ be the support of a maximizer and assume, for contradiction, that there exist $i \in S^\star$ and $j \notin S^\star$ with $u_i < u_j$. Define $S' = S^\star \cup \{j\} \setminus \{i\}$, which

has the same cardinality as $S^\star$. Then

$$F(S') - F(S^\star) = \frac{1}{\sqrt{|S^\star|}}\big(u_j - u_i\big) > 0,$$

contradicting optimality. Therefore $u_i \geq u_j$ whenever $i \in S^\star$ and $j \notin S^\star$. □

## F  THE CONTRASTIVE LOSS IS A DIFFERENTIABLE SURROGATE FOR HR@1

To establish a formal link between the contrastive losses and retrieval, let us consider that a smooth surrogate for optimizing the hit rate @ 1 could be to maximize the probability of the true element being ranked first (i.e., the top-1 probability) (Yang & Koyejo, 2020). According to ListNet (Cao et al., 2007), top-1 probability $P_\mathcal{S}(a)$ of an item $a$ given a set of scores $\mathcal{S}$ is given by:

$$P_\mathcal{S}(a) = \frac{\phi(s_a)}{\sum_{s_j \in \mathcal{S}} \phi(s_j)},$$

where $\phi(\cdot)$ can be any increasing and strictly positive function. When $\phi(\cdot) = \exp(\cdot)$, the top-1 probability is estimated via the softmax operation.

Given a single object as ground truth "positive" molecule (i.e., $\boldsymbol{y}$), the top-one probability can, hence, be maximized by minimizing the following cross-entropy loss $L$:

$$L(\mathcal{S}) = -\log(P_\mathcal{S}(\boldsymbol{y})) = -\log \frac{\exp(s_{\boldsymbol{y}})}{\sum_{s_j \in \mathcal{S}} \exp(s_j)}.$$

Let us now recall that the contrastive loss (4) employs a function $g(\boldsymbol{x}, \boldsymbol{c}_j)$ to obtain a matching score between any input spectrum and candidate structure $\boldsymbol{c}_j \in \mathcal{C}$. Substituting the scores $s$ in the equation above for the function $g$ essentially derives the contrastive loss. As a result, optimizing contrastive losses is equivalent to the retrieval objective of top-1 probability maximization, and, consequently, HR@1 maximization.

The HR@1 and contrastive loss can also be regarded as subset 0/1 accuracy maximization in multi-class classification. In the multi-class classification setting, a surrogate loss for subset 0/1 maximization would be to use traditional categorical cross-entropy to estimate the mode (Dembczyński et al., 2012). Enumerating all possible molecules as classes is computationally infeasible, as argued in Section 2.2. The similarity function inside the contrastive loss allows for efficient computation in an exponential search space with millions of molecules (Cao et al., 2007). Even if one could define all possible molecules as classes, one would have an extremely sparse training signal. Many valid molecules might never appear in your training data, making it hard for the model to learn to predict them. Standard categorical cross-entropy with softmax primarily focuses on maximizing the probability of the correct class (i.e., the entire correct structured output) and minimizing the probability of all incorrect classes. It doesn't inherently understand the internal relationships or similarities between different molecules. However, such a compromise between computational feasibility, expressive power and overfitting mitigation characterizes various types of probabilistic models for structured outputs, such as conditional random fields, probabilistic hierarchical classifiers (Mortier et al., 2021) and autoregressive neural network decoders (Nam et al., 2017).

**On HR@k**  Note that the formal link between the contrastive loss in this study and the retrieval objective is limited to the HR@1 case. Here, we provide some preliminary discussion on the differences between HR@1 and HR@k in terms of (1) Bayes-optimal decisions, and (2) their optimization. To analyze Bayes-optimal decisions for HR@k, the notation in (9) needs to be extended, so that the algorithm returns $k$ molecules instead of one. Then, the result for HR@1 can be immediately extended to HR@k (Yang & Koyejo, 2020). Let us sort the probabilities $p(\boldsymbol{y} \,|\, \boldsymbol{x})$ for all $\boldsymbol{y} \in \mathcal{Y}$ from highest to lowest: $p(\boldsymbol{y}_1 \,|\, \boldsymbol{x}) \geq p(\boldsymbol{y}_2 \,|\, \boldsymbol{x}) \geq \cdots \geq p(\boldsymbol{y}_{|\mathcal{Y}|} \,|\, \boldsymbol{x})$. Then, the Bayes-optimal decision for HR@k is given by the $k$ highest molecules in this list. Furthermore, Yang & Koyejo (2020) showed that top-$k$ calibration is necessary and sufficient for consistency; they prove several hinge-style top-$k$ surrogates proposed in earlier computer-vision work are not top-$k$ calibrated, while giving consistent alternatives. Cross-entropy (in rich function classes) is top-$k$ consistent, but may lose consistency under linear restrictions. These results formalize when utility maximization

(via a surrogate) recovers the Bayes decoder for top-$k$. As a side note, remark that HR@$k$ has also been theoretically analyzed in multi-label retrieval (Menon et al., 2019; Patel et al., 2021), but these results are less relevant for mass spectrometry data.

# G PROOFS FOR REGRET ANALYSIS THEOREMS

For mathematical rigor we first extend the definitions of the utilities from the main text to handle special cases. Let $\mathcal{Y} = \{0,1\}^m$ with $m \geq 2$. For $\boldsymbol{y}, \hat{\boldsymbol{y}} \in \mathcal{Y}$ define:

$$U_{\mathrm{Bit}}(\boldsymbol{y}, \hat{\boldsymbol{y}}) = \frac{1}{m} \sum_{i=1}^{m} \mathbf{1}\{y_i = \hat{y}_i\},$$

$$U_{\mathrm{HR@1}}(\boldsymbol{y}, \hat{\boldsymbol{y}}) = \mathbf{1}\{\boldsymbol{y} = \hat{\boldsymbol{y}}\},$$

$$U_{\mathrm{Tan}}(\boldsymbol{y}, \hat{\boldsymbol{y}}) = \begin{cases} \dfrac{\|\boldsymbol{y} \wedge \hat{\boldsymbol{y}}\|_0}{\|\boldsymbol{y} \vee \hat{\boldsymbol{y}}\|_0}, & \|\boldsymbol{y} \vee \hat{\boldsymbol{y}}\|_0 > 0, \\ 1, & \boldsymbol{y} = \hat{\boldsymbol{y}} = \mathbf{0}, \end{cases}$$

$$U_{\mathrm{Cos}}(\boldsymbol{y}, \hat{\boldsymbol{y}}) = \begin{cases} \dfrac{\boldsymbol{y}^\top \hat{\boldsymbol{y}}}{\|\boldsymbol{y}\| \, \|\hat{\boldsymbol{y}}\|}, & \|\boldsymbol{y}\| \cdot \|\hat{\boldsymbol{y}}\| > 0, \\ 1, & \boldsymbol{y} = \hat{\boldsymbol{y}} = \mathbf{0}, \\ 0, & \text{otherwise.} \end{cases}$$

Here $\| \cdot \|_0$ denotes the $\ell_0$ "norm" and $\| \cdot \|$ the Euclidean norm.

Given a conditional distribution $p(\boldsymbol{y} \mid \boldsymbol{x})$, the Bayes predictor for utility $U$ is defined as

$$\boldsymbol{y}_U^\star(\boldsymbol{x}) \in \arg\max_{\hat{\boldsymbol{y}} \in \mathcal{Y}} \mathbb{E}_{\boldsymbol{Y} \sim p(\cdot|\boldsymbol{x})}\big[U(\boldsymbol{Y}, \hat{\boldsymbol{y}})\big].$$

## G.1 PROOF OF HR@1 DECODING UNDER TO VECTORWISE SIMILARITY

Let denote the expected sim-utility of predicting $\boldsymbol{y}_i$ as:

$$u_i := \mathbb{E}_{\boldsymbol{Y} \sim p(\cdot|\boldsymbol{x})}\big[U_{\mathrm{sim}}(\boldsymbol{y}_i, \boldsymbol{Y})\big] = \sum_{j=1}^{\ell} U_{\mathrm{sim}}(\boldsymbol{y}_i, \boldsymbol{y}_j) p_j.$$

Let $i_\star := \arg\max_i p_i$ be the index of fingerprint from the candidate set $\mathcal{C}$ that has the highest posterior $p_\star = p_{i_\star} = \max_i p_i$. Then let $i_{\mathrm{sim}} := \arg\max_i u_i$ be the index of fingerprint that is sim-optimal (vectorwise) decision, so $\boldsymbol{y}_{\mathrm{sim}}^\star(\boldsymbol{x}) = \boldsymbol{y}_{i_{\mathrm{sim}}}$ and $p_{\mathrm{sim}} := p(\boldsymbol{y}_{\mathrm{sim}}^\star(\boldsymbol{x}) \mid \boldsymbol{x}) = p_{i_{\mathrm{sim}}}$.

**Lemma 1** (Row-wise bounds)**.** *For every* $i \in \{1, 2, \ldots, \ell\}$,

$$p_i + \sigma_{\min}(1 - p_i) \leq u_i \leq p_i + \sigma_{\max}(1 - p_i) = \sigma_{\max} + (1 - \sigma_{\max})p_i. \tag{10}$$

*Proof.* For any index $i \in \{1, \ldots, \ell\}$, by definition

$$u_i = \sum_{j=1}^{\ell} U_{\mathrm{sim}}(\boldsymbol{y}_i, \boldsymbol{y}_j) p_j = U_{\mathrm{sim}}(\boldsymbol{y}_i, \boldsymbol{y}_i) \, p_i + \sum_{j \neq i} U_{\mathrm{sim}}(\boldsymbol{y}_i, \boldsymbol{y}_j) p_j = p_i + \sum_{j \neq i} U_{\mathrm{sim}}(\boldsymbol{y}_i, \boldsymbol{y}_j) p_j. \tag{11}$$

Let $t := \sum_{j \neq i} p_j = 1 - p_i$. If $t = 0$ (equivalently $p_i = 1$) then $u_i = p_i = 1$, and both inequalities in (10) hold with equality. In the case $t > 0$ define normalized nonnegative weights

$$w_j := \frac{p_j}{t}, \qquad j \neq i,$$

so that $\sum_{j \neq i} w_j = 1$ and $w_j \geq 0$ for all $j \neq i$. Using these weights, Eq. (11) becomes

$$u_i = p_i + t \sum_{j \neq i} U_{\mathrm{sim}}(\boldsymbol{y}_i, \boldsymbol{y}_j) \, w_j. \tag{12}$$

By the similarity band, for every $j \neq i$ we have $\sigma_{\min} \leq U_{\mathrm{sim}}(\boldsymbol{y}_i, \boldsymbol{y}_j) \leq \sigma_{\max}$. Therefore, for the convex combination $\sum_{j \neq i} U_{\mathrm{sim}}(\boldsymbol{y}_i, \boldsymbol{y}_j) w_j$ we have the elementary bounds

$$\sigma_{\min} \; \leq \; \sum_{j \neq i} U_{\mathrm{sim}}(\boldsymbol{y}_i, \boldsymbol{y}_j) \, w_j \; \leq \; \sigma_{\max}.$$

Multiplying by $t = 1 - p_i$ and adding $p_i$ to all sides yields

$$p_i + \sigma_{\min}(1 - p_i) \leq p_i + t \sum_{j \neq i} U_{\mathrm{sim}}(\boldsymbol{y}_i, \boldsymbol{y}_j) \, w_j \leq p_i + \sigma_{\max}(1 - p_i) = \sigma_{\max} + (1 - \sigma_{\max}) \, p_i,$$

$\square$

**Theorem 2** (Regret of HR@1 under vectorwise similarity)**.** *Under the additional assumption that the similarity metric is symmetric,* $U_{\mathrm{sim}}(\boldsymbol{y}_i, \boldsymbol{y}_j) = U_{\mathrm{sim}}(\boldsymbol{y}_j, \boldsymbol{y}_i)$, *the regret of HR@1 decoding evaluated under the similarity metric* $U_{\mathrm{sim}}$ *obeys:*

$$\mathcal{R}_{\mathrm{sim}}^{\mathrm{vs\,HR@1}} \leq [(\sigma_{\max} - \sigma_{\min})(1 - p_\star - p_{\mathrm{sim}}) - (1 - \sigma_{\max})(p_\star - p_{\mathrm{sim}})]_+ = \sup_{p(\cdot|\boldsymbol{x})} \mathcal{R}_{\mathrm{sim}}^{\mathrm{vs\,HR@1}},$$

*where* $p_{\mathrm{sim}} = p(\boldsymbol{y}_{\mathrm{sim}}^\star(\boldsymbol{x}) \mid \boldsymbol{x})$ *and* $[z]_+ := \max\{0, z\}$.

*Proof.* The regret can be simply written as

$$\mathcal{R}_{\mathrm{sim}}^{\mathrm{vs\,HR@1}} = \mathbb{E}[U_{\mathrm{sim}}(\boldsymbol{Y}, \boldsymbol{y}_{\mathrm{sim}}^\star(\boldsymbol{x})) \mid \boldsymbol{x}] - \mathbb{E}[U_{\mathrm{sim}}(\boldsymbol{Y}, \boldsymbol{y}_{\mathrm{HR@1}}^\star(\boldsymbol{x})) \mid \boldsymbol{x}] = u_{i_{\mathrm{sim}}} - u_{i_\star}.$$

If $i_{\mathrm{sim}} = i_\star$ there is no regret. Otherwise, by symmetry let $\sigma := U_{\mathrm{sim}}(\boldsymbol{y}_{i_{\mathrm{sim}}}, \boldsymbol{Y}_{i_\star}) = U_{\mathrm{sim}}(\boldsymbol{y}_{i_\star}, \boldsymbol{Y}_{i_{\mathrm{sim}}})$. Using $U_{\mathrm{sim}}(\boldsymbol{y}_i, \boldsymbol{y}_i) = 1$, we expand

$$u_{i_{\mathrm{sim}}} - u_{i_\star} = \sum_{j=1}^\ell U_{\mathrm{sim}}(\boldsymbol{y}_{i_{\mathrm{sim}}}, \boldsymbol{y}_j) p_j - \sum_{j=1}^\ell U_{\mathrm{sim}}(\boldsymbol{y}_{i_\star}, \boldsymbol{y}_j) p_j$$

$$= \left( p_{\mathrm{sim}} + \sigma p_\star + \sum_{j \notin \{i_{\mathrm{sim}}, i_\star\}} U_{\mathrm{sim}}(\boldsymbol{y}_{i_{\mathrm{sim}}}, \boldsymbol{y}_j) p_j \right) - \left( p_\star + \sigma p_{\mathrm{sim}} + \sum_{j \notin \{i_{\mathrm{sim}}, i_\star\}} U_{\mathrm{sim}}(\boldsymbol{y}_{i_\star}, \boldsymbol{y}_j) p_j \right).$$

Bounding the off-diagonal sums using the similarity band yields

$$u_{i_{\mathrm{sim}}} - u_{i_\star} \leq \left( p_{\mathrm{sim}} + \sigma \, p_\star + \sigma_{\max} \sum_{j \notin \{i_{\mathrm{sim}}, i_\star\}} p_j \right) - \left( p_\star + \sigma \, p_{\mathrm{sim}} + \sigma_{\min} \sum_{j \notin \{i_{\mathrm{sim}}, i_\star\}} p_j \right)$$

$$= \sigma(p_\star - p_{\mathrm{sim}}) + (p_{\mathrm{sim}} - p_\star) + (\sigma_{\max} - \sigma_{\min}) \sum_{j \notin \{i_{\mathrm{sim}}, i_\star\}} p_j.$$

Because $\sum_{j \notin \{i_{\mathrm{sim}}, i_\star\}} p_j = 1 - p_{\mathrm{sim}} - p_\star$, we get

$$u_{i_{\mathrm{sim}}} - u_{i_\star} \leq \sigma(p_\star - p_{\mathrm{sim}}) + (p_{\mathrm{sim}} - p_\star) + (\sigma_{\max} - \sigma_{\min})(1 - p_{\mathrm{sim}} - p_\star)$$

$$= \underbrace{[\sigma(p_\star - p_{\mathrm{sim}}) - (p_\star - p_{\mathrm{sim}})]}_{(\sigma - 1)(p_\star - p_{\mathrm{sim}})} + (\sigma_{\max} - \sigma_{\min})(1 - p_{\mathrm{sim}} - p_\star)$$

$$= (\sigma - 1)(p_\star - p_{\mathrm{sim}}) + (\sigma_{\max} - \sigma_{\min})(1 - p_{\mathrm{sim}} - p_\star).$$

Since $p_\star \geq p_{\mathrm{sim}}$, the right-hand side is nondecreasing in $\sigma$, hence it is upper-bounded by replacing $\sigma$ with its maximal allowable value $\sigma_{\max}$:

$$u_{i_{\mathrm{sim}}} - u_{i_\star} \leq (\sigma_{\max} - 1)(p_\star - p_{\mathrm{sim}}) + (\sigma_{\max} - \sigma_{\min})(1 - p_{\mathrm{sim}} - p_\star)$$

$$= -(1 - \sigma_{\max})(p_\star - p_{\mathrm{sim}}) + (\sigma_{\max} - \sigma_{\min})(1 - p_{\mathrm{sim}} - p_\star).$$

Reordering terms,

$$u_{i_{\mathrm{sim}}} - u_{i_\star} \leq (\sigma_{\max} - \sigma_{\min})(1 - p_\star - p_{\mathrm{sim}}) - (1 - \sigma_{\max})(p_\star - p_{\mathrm{sim}})$$

$$= \Big[(\sigma_{\max} - \sigma_{\min})(1 - p_\star - p_{\mathrm{sim}}) - (1 - \sigma_{\max})(p_\star - p_{\mathrm{sim}})\Big].$$

Taking the outer $[\cdot]_+$ to enforce non-negativity of regret yields inequality in Theorem 2.

To prove tightness of achieved upper-bound let us consider following candidate set. Let $i_\star$ and $i_{\text{sim}}$ be two distinct indices. Distribute the remaining mass $1 - p_\star - p_{\text{sim}}$ across the other $\ell - 2$ indices so that each $p_k$ is small (e.g., all equal), which ensures $u_{i_{\text{sim}}} \geq u_k$. Define a candidate set with similarities such that:

$$
\begin{cases}
U_{\text{sim}}(\boldsymbol{y}_i, \boldsymbol{y}_i) = 1 & \text{for all } i, \\
U_{\text{sim}}(\boldsymbol{y}_{i_{\text{sim}}}, \boldsymbol{y}_t) = \sigma_{\max} & \text{for all } t \neq i_{\text{sim}}, \\
U_{\text{sim}}(\boldsymbol{y}_{i_\star}, \boldsymbol{y}_t) = \sigma_{\min} & \text{for all } t \neq i_\star, \\
U_{\text{sim}}(\boldsymbol{y}_k, \boldsymbol{y}_t) = \sigma_{\min} & \text{for all distinct } k, t \notin \{i_\star, i_{\text{sim}}\}.
\end{cases}
$$

Note that symmetry forces $U_{\text{sim}}(\boldsymbol{y}_{i_\star}, \boldsymbol{y}_{i_{\text{sim}}}) = U_{\text{sim}}(\boldsymbol{y}_{i_{\text{sim}}}, \boldsymbol{y}_{i_\star}) = \sigma_{\max}$. Now with these similarities, we show that the regret is equal to the upper-bound.

$$
\begin{aligned}
u_{i_{\text{sim}}} - u_{i_\star} &= \sum_{j=1}^{\ell} U_{\text{sim}}(\boldsymbol{y}_{i_{\text{sim}}}, \boldsymbol{y}_j) p_j - \sum_{j=1}^{\ell} U_{\text{sim}}(\boldsymbol{y}_{i_\star}, \boldsymbol{y}_j) p_j \\
&= \left( p_{\text{sim}} + U_{\text{sim}}(\boldsymbol{y}_{i_{\text{sim}}}, \boldsymbol{y}_{i_\star}) p_\star + \sum_{j \notin \{i_{\text{sim}}, i_\star\}} U_{\text{sim}}(\boldsymbol{y}_{i_{\text{sim}}}, \boldsymbol{y}_j) p_j \right) \\
&\quad - \left( p_\star + U_{\text{sim}}(\boldsymbol{y}_{i_\star}, \boldsymbol{y}_{i_{\text{sim}}}) p_{\text{sim}} + \sum_{j \notin \{i_{\text{sim}}, i_\star\}} U_{\text{sim}}(\boldsymbol{y}_{i_\star}, \boldsymbol{y}_j) p_j \right) \\
&= \left( p_{\text{sim}} + \sigma_{\max} p_\star + \sigma_{\max} \sum_{t \notin \{i_{\text{sim}}, i_\star\}} p_t \right) - \left( p_\star + \sigma_{\max} p_{\text{sim}} + \sigma_{\min} \sum_{t \notin \{i_{\text{sim}}, i_\star\}} p_t \right) \\
&= \sigma_{\max}(p_\star - p_{\text{sim}}) + (p_{\text{sim}} - p_\star) + (\sigma_{\max} - \sigma_{\min}) \sum_{t \notin \{i_{\text{sim}}, i_\star\}} p_t \\
&= \sigma_{\max}(p_\star - p_{\text{sim}}) + (p_{\text{sim}} - p_\star) + (\sigma_{\max} - \sigma_{\min})(1 - p_\star - p_{\text{sim}}) \\
&= (\sigma_{\max} - \sigma_{\min})(1 - p_\star - p_{\text{sim}}) - (1 - \sigma_{\max})(p_\star - p_{\text{sim}}),
\end{aligned}
$$

which is exactly the upper-bound (without the $[\cdot]_+$).

$\square$

### G.2 Proof of vectorwise similarity decoding under HR@1

**Theorem 3** (Regret of vectorwise similarity under HR@1).

$$
\mathcal{R}_{\text{HR@1}}^{\text{vs sim}} \leq \min \left\{ p_\star, \frac{\sigma_{\max} - \sigma_{\min}}{1 - \sigma_{\max}} (1 - p_\star) \right\}.
$$

*Proof.* By optimality, $u_{i_{\text{sim}}} \geq u_{i_\star}$. We apply the row-wise *upper* bound Eq. (10) to $u_{i_{\text{sim}}}$ and the *lower* bound to $u_{i_\star}$:

$$
\begin{aligned}
u_{i_{\text{sim}}} &\leq \sigma_{\max} + (1 - \sigma_{\max}) \, p_{i_{\text{sim}}}, \\
u_{i_\star} &\geq p_\star + \sigma_{\min}(1 - p_\star) = p_\star + \sigma_{\min} - \sigma_{\min} p_\star.
\end{aligned}
$$

Since $u_{i_{\text{sim}}} \geq u_{i_\star}$, we have

$$
\begin{aligned}
\sigma_{\max} + (1 - \sigma_{\max}) \, p_{i_{\text{sim}}} &\geq p_\star + \sigma_{\min}(1 - p_\star) \\
&= p_\star + \sigma_{\min} - \sigma_{\min} p_\star.
\end{aligned}
$$

Subtracting $\sigma_{\max}$ from both sides gives

$$
(1 - \sigma_{\max}) \, p_{i_{\text{sim}}} \geq p_\star + \sigma_{\min} - \sigma_{\min} p_\star - \sigma_{\max}.
$$

Rewriting the right-hand side by grouping the $p_\star$ terms,

$$p_\star + \sigma_{\min} - \sigma_{\min}p_\star - \sigma_{\max} = p_\star(1 - \sigma_{\min}) + \sigma_{\min} - \sigma_{\max}$$
$$= p_\star(1 - \sigma_{\min}) - (\sigma_{\max} - \sigma_{\min}).$$

Hence

$$(1 - \sigma_{\max})\, p_{i_{\text{sim}}} \ \geq \ p_\star(1 - \sigma_{\min}) - (\sigma_{\max} - \sigma_{\min}).$$

Since $1 - \sigma_{\max} > 0$ (off-diagonal similarities are strictly $< 1$), we can divide both sides by $1 - \sigma_{\max}$ to get:

$$p_{i_{\text{sim}}} \ \geq \ \frac{p_\star(1 - \sigma_{\min}) - (\sigma_{\max} - \sigma_{\min})}{1 - \sigma_{\max}}.$$

Therefore,

$$p_\star - p_{i_{\text{sim}}} \leq p_\star - \frac{p_\star(1 - \sigma_{\min}) - (\sigma_{\max} - \sigma_{\min})}{1 - \sigma_{\max}}$$
$$= \frac{p_\star(1 - \sigma_{\max})}{1 - \sigma_{\max}} - \frac{p_\star(1 - \sigma_{\min}) - (\sigma_{\max} - \sigma_{\min})}{1 - \sigma_{\max}}$$
$$= \frac{p_\star(1 - \sigma_{\max}) - p_\star(1 - \sigma_{\min}) + (\sigma_{\max} - \sigma_{\min})}{1 - \sigma_{\max}}$$
$$= \frac{-p_\star(\sigma_{\max} - \sigma_{\min}) + (\sigma_{\max} - \sigma_{\min})}{1 - \sigma_{\max}}$$
$$= \frac{(\sigma_{\max} - \sigma_{\min})(1 - p_\star)}{1 - \sigma_{\max}}.$$

Together with the trivial bound $p_\star - p_{i_{\text{sim}}} \leq p_\star$, this proves Theorem 3. $\qquad\square$

### G.3 A SUFFICIENT CONDITION FOR NO REGRET AGREEMENT OF DECISIONS

**Theorem 5** (Agreement of top-1 and vector-wise choices). *Let $\Delta := p_{i_\star} - \max_{j \neq i_\star} p_j \in [0, p_{i_\star}]$ be the margin between the highest and the second highest probability. Then $i_{\text{sim}} = i_\star$; i.e., HR@1 and the* sim-*optimal decisions coincide if:*

$$\Delta \ \geq \ \frac{\sigma_{\max} - \sigma_{\min}}{1 - \sigma_{\max}}\,(1 - p_\star). \tag{13}$$

*Proof.* Assume the agreement condition (13) holds. Because under our assumption on the similarity band we have $1 - \sigma_{\max} \geq 0$, multiplying both sides of (13) by $1 - \sigma_{\max}$ gives:

$$(1 - \sigma_{\max})\Delta \geq (\sigma_{\max} - \sigma_{\min})(1 - p_\star)$$
$$0 \geq (\sigma_{\max} - \sigma_{\min})(1 - p_\star) - (1 - \sigma_{\max})\Delta\,.$$

Let $p_{(2)} := \max_{j \neq i_\star} p_j$ be the second highest conditional probability. By definition of the margin $\Delta$ we have $p_{(2)} = p_\star - \Delta$. Fix any $j \neq i_\star$. By the row-wise bounds:

$$u_{i_\star} = \sum_{k=1}^{\ell} U_{\text{sim}}(\boldsymbol{y}_{i_\star}, \boldsymbol{y}_k)p_k \geq \ p_\star + \sigma_{\min}(1 - p_\star)$$

$$u_j = \sum_{k=1}^{\ell} U_{\text{sim}}(\boldsymbol{y}_j, \boldsymbol{y}_k)p_k \leq \sigma_{\max} + (1 - \sigma_{\max})p_j\,.$$

Therefore

$$u_{i_\star} - u_j \geq p_\star + \sigma_{\min}(1 - p_\star) - \Big(\sigma_{\max} + (1 - \sigma_{\max})\,p_j\Big)$$
$$\geq p_\star + \sigma_{\min}(1 - p_\star) - \Big(\sigma_{\max} + (1 - \sigma_{\max})(p_\star - \Delta)\Big)$$
$$= -(\sigma_{\max} - \sigma_{\min})(1 - p_\star) + (1 - \sigma_{\max})\Delta.$$

If (13) holds, then the right-hand side is nonnegative, so $u_{i_\star} \geq u_j$ for every $j \neq i_\star$. Hence $i_\star \in \arg\max_i u_i$ and the two decoders agree. $\qquad\square$

## G.4 Worst-case regrets depending on $m$

**Theorem 4** (Regret bitwise-loss under vectorwise similarity)**.**

$$\sup_{p(\cdot|\boldsymbol{x})} \mathcal{R}_{\mathrm{Tan}}^{\mathrm{vs\,Bit}} \geq \tfrac{1}{2}, \qquad \sup_{p(\cdot|\boldsymbol{x})} \mathcal{R}_{\mathrm{Cos}}^{\mathrm{vs\,Bit}} \geq \tfrac{1}{2} \quad (m \geq 3).$$

*Proof.* **(a) HR@1.)** The equality $\sup \mathcal{R}_{\mathrm{HR@1}}^{\mathrm{vs\,Bit}} = \tfrac{1}{2}$ is a known tight result; see, e.g., the analysis of subset accuracy regret when predicting with Hamming-optimal rules in Dembczyński et al. (2012).

**(b) Tanimoto and (c) Cosine: lower bounds $\geq \tfrac{1}{2}$.** Fix $\varepsilon > 0$ small and define the following conditional distribution on $\mathcal{Y}$:

$$p_\varepsilon(\boldsymbol{y} \mid \boldsymbol{x}) = \begin{cases} 0.5 - \varepsilon, & \text{if } \boldsymbol{y} = \boldsymbol{y}^1, \\ 0.5 - (2m-3)\varepsilon, & \text{if } \boldsymbol{y} = \bar{\boldsymbol{y}}^1, \\ 2\varepsilon, & \text{if } y_1 = 0 \text{ and } d_H(\boldsymbol{y}, \bar{\boldsymbol{y}}^1) = 1, \\ 0, & \text{otherwise,} \end{cases}$$

where $\boldsymbol{y}^1 = (1, 0, \ldots, 0)$, $\bar{\boldsymbol{y}}^1 = (0, 1, \ldots, 1)$, and $d_H$ is Hamming distance. For $m \geq 3$ this is a valid distribution (nonnegative and summing to 1) and every $\boldsymbol{y}$ in its support has $\|\boldsymbol{y}\| > 0$.

*Step 1: The Hamming-optimal predictor is the zero vector.* Let $p_\varepsilon^{(i)}(1) := \mathbb{P}(Y_i = 1 \mid \boldsymbol{x})$ under $p_\varepsilon$. Then

$$p_\varepsilon^{(1)}(1) = 0.5 - \varepsilon, \qquad p_\varepsilon^{(j)}(1) = (0.5 - (2m-3)\varepsilon) + (m-2) \cdot 2\varepsilon = 0.5 - \varepsilon \quad (j \geq 2).$$

Thus $p_\varepsilon^{(i)}(0) = 0.5 + \varepsilon$ for all $i$, and the Bayes rule for $U_{\mathrm{Bit}}$ is

$$\boldsymbol{y}_{\mathrm{Bit}}^\star(\boldsymbol{x}) = \boldsymbol{0}.$$

*Step 2: A candidate predictor with large expected utility.* Consider $\hat{\boldsymbol{y}} = \bar{\boldsymbol{y}}^1$.

Tanimoto. For $\boldsymbol{y} = \boldsymbol{y}^1$ the intersection is 0, so $U_{\mathrm{Tan}}(\boldsymbol{y}^1, \hat{\boldsymbol{y}}) = 0$. For $\boldsymbol{y} = \bar{\boldsymbol{y}}^1$ we have $U_{\mathrm{Tan}} = 1$. If $\boldsymbol{y}$ is a neighbor of $\bar{\boldsymbol{y}}^1$ with $y_1 = 0$ and $d_H(\boldsymbol{y}, \bar{\boldsymbol{y}}^1) = 1$, then $\|\boldsymbol{y} \wedge \hat{\boldsymbol{y}}\|_0 = m - 2$ and $\|\boldsymbol{y} \vee \hat{\boldsymbol{y}}\|_0 = m - 1$, hence $U_{\mathrm{Tan}} = (m-2)/(m-1)$. Therefore

$$\mathbb{E}\big[U_{\mathrm{Tan}}(\boldsymbol{Y}, \hat{\boldsymbol{y}}) \mid \boldsymbol{x}\big] = 0 \cdot \big(0.5 - \varepsilon\big) + 1 \cdot \big(0.5 - (2m-3)\varepsilon\big) + (m-1) \cdot 2\varepsilon \cdot \frac{m-2}{m-1}$$

$$= 0.5 - \varepsilon.$$

Since $U_{\mathrm{Tan}}(\boldsymbol{y}, \boldsymbol{0}) = 0$ for all $\boldsymbol{y} \neq \boldsymbol{0}$, we have

$$\mathbb{E}\big[U_{\mathrm{Tan}}(\boldsymbol{Y}, \boldsymbol{y}_{\mathrm{Bit}}^\star(\boldsymbol{x})) \mid \boldsymbol{x}\big] = 0,$$

and consequently

$$\mathbb{E}\big[U_{\mathrm{Tan}}(\boldsymbol{Y}, \hat{\boldsymbol{y}}) \mid \boldsymbol{x}\big] - \mathbb{E}\big[U_{\mathrm{Tan}}(\boldsymbol{Y}, \boldsymbol{y}_{\mathrm{Bit}}^\star(\boldsymbol{x})) \mid \boldsymbol{x}\big] = 0.5 - \varepsilon.$$

Since $h_{\mathrm{Tan}}^\star$ maximizes expected Tanimoto, its performance is at least that of $\hat{\boldsymbol{y}}$, hence $\sup \mathcal{R}_{\mathrm{Tan}} \geq 0.5 - \varepsilon$. Letting $\varepsilon \downarrow 0$ yields $\sup \mathcal{R}_{\mathrm{Tan}}^{\mathrm{vs\,Bit}} \geq \tfrac{1}{2}$.

Cosine. With $\hat{\boldsymbol{y}} = \bar{\boldsymbol{y}}^1$ we have:

$$U_{\mathrm{Cos}}(\boldsymbol{y}^1, \hat{\boldsymbol{y}}) = 0, \qquad U_{\mathrm{Cos}}(\bar{\boldsymbol{y}}^1, \hat{\boldsymbol{y}}) = 1, \qquad U_{\mathrm{Cos}}(\boldsymbol{y}, \hat{\boldsymbol{y}}) = \frac{m-2}{\sqrt{m-2}\sqrt{m-1}} = \sqrt{\frac{m-2}{m-1}}$$

for any neighbor $\boldsymbol{y}$ of $\bar{\boldsymbol{y}}^1$ as above. Therefore

$$\mathbb{E}\big[U_{\mathrm{Cos}}(\boldsymbol{Y}, \hat{\boldsymbol{y}}) \mid \boldsymbol{x}\big] = 0 \cdot \big(0.5 - \varepsilon\big) + 1 \cdot \big(0.5 - (2m-3)\varepsilon\big) + (m-1) \cdot 2\varepsilon \cdot \sqrt{\frac{m-2}{m-1}}$$

$$= 0.5 + \varepsilon \left[ -(2m-3) + 2(m-1)\sqrt{1 - \frac{1}{m-1}} \right].$$

Since $\sqrt{1-\alpha} \geq 1 - \alpha$ for $\alpha \in [0,1]$, the bracketed term is $\geq -1$, hence

$$\mathbb{E}\big[U_{\mathrm{Cos}}(\boldsymbol{Y}, \hat{\boldsymbol{y}}) \mid \boldsymbol{x}\big] \;\geq\; 0.5 - \varepsilon.$$

Again $U_{\mathrm{Cos}}(\boldsymbol{y}, \boldsymbol{0}) = 0$ for all $\boldsymbol{y}$, so

$$\mathbb{E}\big[U_{\mathrm{Cos}}(\boldsymbol{Y}, \boldsymbol{y}_{\mathrm{Bit}}^{\star}(\boldsymbol{x})) \mid \boldsymbol{x}\big] = 0,$$

and therefore

$$\mathbb{E}\big[U_{\mathrm{Cos}}(\boldsymbol{Y}, \boldsymbol{y}_{\mathrm{Cos}}^{\star}(\boldsymbol{x})) \mid \boldsymbol{x}\big] - \mathbb{E}\big[U_{\mathrm{Cos}}(\boldsymbol{Y}, \boldsymbol{y}_{\mathrm{Bit}}^{\star}(\boldsymbol{x})) \mid \boldsymbol{x}\big] \;\geq\; 0.5 - \varepsilon.$$

Letting $\varepsilon \downarrow 0$ gives $\sup \mathcal{R}_{\mathrm{Cos}}^{\mathrm{vs\,Bit}} \geq \frac{1}{2}$. $\qquad\square$

Remarks: (i) The equality $\sup \mathcal{R}_{\mathrm{HR@1}}^{\mathrm{vs\,Bit}} = \frac{1}{2}$ is tight; see Dembczyński et al. (2012). For Tanimoto and Cosine, the bounds are *lower* bounds obtained by explicit constructions; sharper (possibly tight) constants are an interesting open direction. (ii) The proof uses the convention $U_{\mathrm{Cos}}(\boldsymbol{y}, \boldsymbol{0}) = 0$ and similarly for $U_{\mathrm{Tan}}$; the construction avoids $\boldsymbol{y} = \boldsymbol{0}$ so denominators remain well-defined.

**Theorem 6** (Worst-case regret when decoding by HR@1). *For $m > 3$ it holds that:*

$$22 \qquad \sup_{p(\cdot|\boldsymbol{x})} \mathcal{R}_{\mathrm{Tan}}^{\mathrm{vs\,HR@1}} \;\geq\; \frac{m-1}{m+2}, \qquad \sup_{p(\cdot|\boldsymbol{x})} \mathcal{R}_{\mathrm{Cos}}^{\mathrm{vs\,HR@1}} \;\geq\; \frac{\sqrt{m(m-1)}}{m+2}.$$

*Proof.* Fix $\varepsilon > 0$ small and let, for $\alpha := \frac{1}{m+2} - \varepsilon$,

$$p_{\varepsilon}(\boldsymbol{y} \mid \boldsymbol{x}) = \begin{cases} \frac{1}{m+2} + (m+1)\varepsilon, & \boldsymbol{y} = \boldsymbol{0}_m, \\ \alpha, & \boldsymbol{y} = \boldsymbol{1}_m \ \text{or} \ d_H(\boldsymbol{y}, \boldsymbol{0}_m) = m - 1, \\ 0, & \text{otherwise}, \end{cases}$$

where $d_H$ is Hamming distance. This is a valid distribution since the support has $(m+1)$ points with mass $\alpha$ and one point with mass $\frac{1}{m+2} + (m+1)\varepsilon$, summing to 1.

**Step 1: The decoders $\boldsymbol{y}_{\mathrm{HR@1}}^{\star}$ and $\boldsymbol{y}_{\mathrm{Bit}}^{\star}$ under $p_{\varepsilon}$.** The joint mode is $\boldsymbol{0}_m$ because $p_{\varepsilon}(\boldsymbol{0}_m) = \frac{1}{m+2} + (m+1)\varepsilon > \alpha = p_{\varepsilon}(\boldsymbol{y})$ for every other support point. Hence $\boldsymbol{y}_{\mathrm{HR@1}}^{\star}(\boldsymbol{x}) = \boldsymbol{0}_m$. For marginals,

$$\mathbb{P}(Y_i = 1 \mid \boldsymbol{x}) = p_{\varepsilon}(\boldsymbol{1}_m) + (m-1)\, p_{\varepsilon}(\text{``exactly one zero not at } i\text{''}) = \alpha + (m-1)\alpha = m\alpha,$$

so $\mathbb{P}(Y_i = 0 \mid \boldsymbol{x}) = 1 - m\alpha = \frac{2}{m+2} + m\varepsilon$. For $m \geq 3$ and $\varepsilon$ sufficiently small, $m\alpha > \frac{1}{2}$, thus $\boldsymbol{y}_{\mathrm{Bit}}^{\star}(\boldsymbol{x}) = \boldsymbol{1}_m$.

**Step 2: Bitwise accuracy (tight equality).** Since $U_{\mathrm{Bit}}(\boldsymbol{y}, \boldsymbol{1}_m) = \frac{\|\boldsymbol{y}\|_0}{m}$ and $U_{\mathrm{Bit}}(\boldsymbol{y}, \boldsymbol{0}_m) = 1 - \frac{\|\boldsymbol{y}\|_0}{m}$,

$$\mathbb{E}\big[U_{\mathrm{Bit}}(\boldsymbol{Y}, \boldsymbol{1}_m) \mid \boldsymbol{x}\big] = m\alpha, \qquad \mathbb{E}\big[U_{\mathrm{Bit}}(\boldsymbol{Y}, \boldsymbol{0}_m) \mid \boldsymbol{x}\big] = 1 - m\alpha.$$

Thus the (pointwise) regret incurred by HR@1 equals

$$m\alpha - (1 - m\alpha) = 2m\alpha - 1 = \frac{m-2}{m+2} - 2m\varepsilon.$$

Letting $\varepsilon \downarrow 0$ shows $\sup \mathcal{R}_{\mathrm{Bit}}^{\mathrm{vs\,HR@1}} \geq \frac{m-2}{m+2}$. The reverse inequality (hence equality) is known to be tight; see Dembczyński et al. (2012).

**Step 3: Tanimoto (lower bound).** Under the conventions in the definition, $U_{\mathrm{Tan}}(\boldsymbol{0}_m, \boldsymbol{0}_m) = 1$ and $U_{\mathrm{Tan}}(\boldsymbol{y}, \boldsymbol{0}_m) = 0$ for $\boldsymbol{y} \neq \boldsymbol{0}_m$, so

$$\mathbb{E}\big[U_{\mathrm{Tan}}(\boldsymbol{Y}, \boldsymbol{y}_{\mathrm{HR@1}}^{\star}) \mid \boldsymbol{x}\big] = p_{\varepsilon}(\boldsymbol{0}_m) = \frac{1}{m+2} + (m+1)\varepsilon.$$

Consider $\hat{\boldsymbol{y}} = \boldsymbol{1}_m$. Then $U_{\mathrm{Tan}}(\boldsymbol{1}_m, \hat{\boldsymbol{y}}) = 1$ and, for any $\boldsymbol{y}$ with exactly one zero, $U_{\mathrm{Tan}}(\boldsymbol{y}, \hat{\boldsymbol{y}}) = \frac{m-1}{m}$, while $U_{\mathrm{Tan}}(\boldsymbol{0}_m, \hat{\boldsymbol{y}}) = 0$. Hence

$$\mathbb{E}\big[U_{\mathrm{Tan}}(\boldsymbol{Y}, \hat{\boldsymbol{y}}) \mid \boldsymbol{x}\big] = \alpha \cdot 1 + m\alpha \cdot \frac{m-1}{m} = m\alpha = \frac{m}{m+2} - m\varepsilon.$$

Since $\boldsymbol{y}_{\mathrm{Tan}}^{\star}$ maximizes expected Tanimoto,

$$\sup_{p(\cdot|\boldsymbol{x})} \mathcal{R}_{\mathrm{Tan}}^{\mathrm{vs\,HR@1}} \ \geq \ m\alpha - \left(\tfrac{1}{m+2} + (m+1)\varepsilon\right) = \frac{m-1}{m+2} - (2m+1)\varepsilon.$$

Let $\varepsilon \downarrow 0$ to obtain $\sup \mathcal{R}_{\mathrm{Tan}}^{\mathrm{vs\,HR@1}} \geq \frac{m-1}{m+2}$.

**Step 4: Cosine (lower bound).** With $\boldsymbol{y}_{\mathrm{HR@1}}^{\star} = \mathbf{0}_m$ and our convention $U_{\mathrm{Cos}}(\mathbf{0}_m, \mathbf{0}_m) = 1$ and 0 otherwise,

$$\mathbb{E}\big[U_{\mathrm{Cos}}(\boldsymbol{Y}, \boldsymbol{y}_{\mathrm{HR@1}}^{\star}) \mid \boldsymbol{x}\big] = p_{\varepsilon}(\mathbf{0}_m) = \tfrac{1}{m+2} + (m+1)\varepsilon.$$

Take again $\hat{\boldsymbol{y}} = \mathbf{1}_m$. Then $U_{\mathrm{Cos}}(\mathbf{1}_m, \hat{\boldsymbol{y}}) = 1$ and, for any $\boldsymbol{y}$ with exactly one zero,

$$U_{\mathrm{Cos}}(\boldsymbol{y}, \hat{\boldsymbol{y}}) = \frac{m-1}{\sqrt{m}\sqrt{m-1}} = \sqrt{\frac{m-1}{m}}.$$

Therefore

$$\mathbb{E}\big[U_{\mathrm{Cos}}(\boldsymbol{Y}, \hat{\boldsymbol{y}}) \mid \boldsymbol{x}\big] = \alpha \cdot 1 + m\alpha \cdot \sqrt{\frac{m-1}{m}} = \alpha\Big(1 + \sqrt{m(m-1)}\Big).$$

By optimality of $\boldsymbol{y}_{\mathrm{Cos}}^{\star}$,

$$\sup_{p(\cdot|\boldsymbol{x})} \mathcal{R}_{\mathrm{Cos}}^{\mathrm{vs\,HR@1}} \ \geq \ \alpha\Big(1 + \sqrt{m(m-1)}\Big) - \Big(\tfrac{1}{m+2} + (m+1)\varepsilon\Big) = \frac{\sqrt{m(m-1)}}{m+2} - \Big(\sqrt{m(m-1)} + m + 2\Big)\varepsilon.$$

Letting $\varepsilon \downarrow 0$ yields $\sup \mathcal{R}_{\mathrm{Cos}}^{\mathrm{vs\,HR@1}} \geq \frac{\sqrt{m(m-1)}}{m+2}$, as claimed. $\qquad\square$

Remarks: (i) The equality for $U_{\mathrm{Bit}}$ is tight; see Dembczyński et al. (2012). (ii) The lower bounds for Tanimoto and Cosine come from an explicit adversarial family $p_{\varepsilon}$; sharper constants (or tightness) are an open question. (iii) If one instead sets $U_{\mathrm{Tan}}(\mathbf{0}, \mathbf{0}) = 0$ and/or $U_{\mathrm{Cos}}(\mathbf{0}, \mathbf{0}) = 0$, the bounds shift by $-\frac{1}{m+2}$ accordingly; we adopted the "empty equals empty" similarity convention to match common Jaccard practice and to align the constants with the statement above. (iv) For $m = 2$, the construction yields $\boldsymbol{y}_{\mathrm{Bit}} = \mathbf{0}$ as well, and the bitwise gap collapses to $0 = (m-2)/(m+2)$, consistent with the theorem.

## H  How theory informs fingerprint selection

All previous empirical results are presented using Morgan ($r = 2$, 4096-bit) fingerprints, because they constitute the *de facto* standard in metabolomics (Goldman et al., 2023; Bushuiev et al., 2024a). However, all results in Section 4.2 abstract away from specific fingerprint choice. Instead, they rely on a notion of "similarity bands", meaning that different binary representations of molecules naturally fit into the same framework.

To showcase how our theory may inform fingerprint selection, consider that minimizing regret is favorable, as in that case, improving fingerprint similarity predictions is expected to improve retrieval. The regrets formulated in Theorem 2 and Theorem 3 are contingent upon (1) conditional probabilities $p(\boldsymbol{y} \mid \boldsymbol{x})$, and (2) $\sigma_{\min}$ and $\sigma_{\max}$. While the former are unknown, the latter can be computed from data: they are the minimum and maximum similar negative candidates w.r.t. the true molecule, respectively. Our theorems show that regret increases with $\sigma_{\max}$ and $\sigma_{\max} - \sigma_{\min}$. Computing these values for any similarity function and/or type of fingerprint may, hence, serve model design toward minimizing regret.

Here, we perform experiment with five fingerprint alternatives to Morgan ($r = 2$), namely: (1) Morgan ($r = 4$), (2) Morgan ($r = 6$), (3) Morgan ($r = 8$), (4) RDKit fingerprints (Landrum, 2013), and (5) MAP4 fingerprints (Capecchi et al., 2020). All of these fingerprints are computed in a way such that they are binary and comprise a 4096-length bitvector. To estimate how well these representations are fit to minimize regret, we compute empirical distributions of $\sigma_{\min}$, $\sigma_{\max}$, and $\sigma_{\max} - \sigma_{\min}$, as in Figure 5. From Table 6, one can see that Morgan fingerprints with increasing radii and MAP4 have favorable distributions in how "discriminative" similarities between candidates can be. For RDKit fingerprints, on the other hand, there are often negative candidates with high ($> 0.5$) similarity to the true molecule.

To experimentally validate to what extent the above-mentioned fingerprint types minimize regret, we optimized IoU loss and Contrastive (Emb-Cos) models using these fingerprints as targets/inputs. As discussed previously, these two loss functions serve as surrogates for Tanimoto similarity maximization and HR@1 maximization, respectively. For all of these configurations, models were run five times each in exactly the same configuration as for the main experiments (also using the same tuned learning rates). The resulting performances (Table 7) confirm our theory: the difference between HR@1 using both IoU loss and contrastive losses broadly correlates with values of $\sigma_{\min}$ and $\sigma_{\max}$ across fingerprint types. Performance-

Table 6: Empirical medians of $\sigma_{\min}$, $\sigma_{\max}$, and $\sigma_{\max} - \sigma_{\min}$ distributions in MassSpec-Gym equal-mass candidate sets using Tanimoto similarity, depending on the type of fingerprint.

| Fingerprint type | $\sigma_{\min}$ | $\sigma_{\max}$ | $\sigma_{\max} - \sigma_{\min}$ |
|---|---|---|---|
| Morgan $r = 2$ | 0.026 | 0.367 | 0.342 |
| Morgan $r = 4$ | 0.018 | 0.241 | 0.223 |
| Morgan $r = 6$ | 0.017 | 0.211 | 0.193 |
| Morgan $r = 8$ | 0.017 | 0.204 | 0.187 |
| RDKit | 0.039 | 0.512 | 0.471 |
| MAP4 | 0.025 | 0.198 | 0.173 |

wise, the story is more nuanced: while Morgan fingerprints with increasing radii showed favorable $\sigma_{\min}$ and $\sigma_{\max}$ distributions, their hit rates using both IoU loss and contrastive losses progressively worsen with radius. An explanation may be found in their average Tanimoto similarity scores, which similarly worsen with radius. While comparing these scores between different types of targets should be performed with care, they do show that increasing the Morgan fingerprint radius makes for a more difficult prediction target. It is not unreasonable to assume that, in this case, conditional probabilities $p(\boldsymbol{y} \mid \boldsymbol{x})$ in general follow a more uniform distribution. This similarly inflates the regrets in Theorem 2 and Theorem 3 through the $(1 - p^*)$ terms. As such, regret is additionally influenced by how easy it is to predict any given target fingerprint.

Table 7: Fingerprint benchmark results. Retrieval scores are those using equal mass candidates. All models used in this table were trained with a learning rate of 7e-5. Numbers report averages and standard deviations over 5 model runs. The best and second-best performing models for each metric are indicated with boldface and underlined, respectively.

| Fingerprint type | Scores obtained with IoU loss | | | | Scores obtained with Contr. Emb-Cos loss | | |
|---|---|---|---|---|---|---|---|
| | Tanimoto[1] | HR@1 | HR@5 | HR@20 | HR@1 | HR@5 | HR@20 |
| Morgan $r = 2$ | $19.42 \pm 0.13$ | $5.58 \pm 0.32$ | $13.01 \pm 0.39$ | $26.51 \pm 0.50$ | $\mathbf{12.29 \pm 0.69}$ | $\mathbf{26.09 \pm 1.43}$ | $\underline{45.05 \pm 0.71}$ |
| Morgan $r = 4$ | $13.61 \pm 0.02$ | $5.56 \pm 0.48$ | $12.83 \pm 0.64$ | $25.80 \pm 0.66$ | $10.55 \pm 0.54$ | $23.96 \pm 1.03$ | $41.41 \pm 0.92$ |
| Morgan $r = 6$ | $11.92 \pm 0.22$ | $5.65 \pm 0.27$ | $13.27 \pm 0.26$ | $25.54 \pm 0.56$ | $09.51 \pm 0.67$ | $21.18 \pm 0.84$ | $37.54 \pm 0.75$ |
| Morgan $r = 8$ | $11.25 \pm 0.09$ | $5.68 \pm 0.37$ | $12.88 \pm 0.52$ | $25.81 \pm 0.51$ | $08.72 \pm 0.52$ | $19.67 \pm 0.81$ | $36.63 \pm 0.77$ |
| RDKit | $\mathbf{30.32 \pm 0.15}$ | $\underline{5.78 \pm 0.17}$ | $15.00 \pm 0.40$ | $\mathbf{29.96 \pm 0.62}$ | $09.63 \pm 0.61$ | $22.49 \pm 0.60$ | $44.13 \pm 1.54$ |
| MAP4 | $\underline{20.48 \pm 0.15}$ | $\mathbf{7.66 \pm 0.26}$ | $\mathbf{15.67 \pm 0.18}$ | $\underline{29.41 \pm 0.21}$ | $\underline{10.95 \pm 0.83}$ | $\underline{24.89 \pm 1.39}$ | $\mathbf{45.19 \pm 1.29}$ |

[1]: Comparing Tanimoto scores between fingerprint types should be performed with care, as they concern different targets. If anything, these scores reflect how "predictable" a given fingerprint type is.

The best retrieval scores using an IoU loss model were obtained by predicting either RDKit or MAP4 fingerprints (Table 7). For MAP4 fingerprints especially, retrieval models comparatively perform well using both IoU loss or contrastive loss. Compared to Morgan ($r = 2$) fingerprints, MAP4 fingerprints are more favorable in terms of similarity band distributions (Table 6), but display comparable Tanimoto similarity scores. It is not unreasonable to assume that the latter demonstrates their conditional probabilities $p(\boldsymbol{y} \mid \boldsymbol{x})$ may be similarly distributed. As such, MAP4 fingerprints should have have smaller regrets between optimal HR@1- and optimal Tanimoto similarity decisions. Table Table 6 demonstrates this to be the case, as HR@1 are closer to each other for MAP4 results as they are for results using Morgan ($r = 2$) fingerprints. Our theory and experiments, hence, show that MAP4 fingerprints may serve as a potentially superior or complementary fingerprint target to the more conventional Morgan fingerprints.

## I FROM BAYES REGRET TO FINITE-SAMPLE PREDICTORS

The regret analysis in Section 4.2 is carried out at the *Bayes level*: we compare decision rules that are optimal under different utilities $U$ and $V$ (e.g. HR@1, Tanimoto, cosine, bitwise accuracy). In practice, however, fingerprint predictors are trained on a *finite* sample by minimizing an empirical loss. The resulting decision rule therefore deviates from the Bayes-optimal rule.

In this section we separate these two phenomena:

- a purely *intrinsic Bayes-level mismatch* between using $U$-optimal or $V$-optimal decisions (captured by the regret bounds in Section 4.2 and Appendix G), and
- a *finite-sample generalization component* arising from the estimation error of the fingerprint predictor trained by empirical risk minimization (ERM).

Formally, we obtain an exact decomposition of the $U$-regret into a Bayes term plus a residual term, and we show how the residual can be controlled via standard learning-theoretic bounds on the surrogate loss.

We recall the probabilistic framework from Section 4.1. Let $\mathcal{F}$ be a hypothesis class of fingerprint predictors $f : \mathcal{X} \to [0,1]^m$, for instance the neural networks of the form $f(\boldsymbol{x}) = \sigma(\boldsymbol{W}_{\text{out}}E(\boldsymbol{x}))$ described in Appendix A. Training is performed by minimizing an empirical loss

$$\widehat{L}_n(f) := \frac{1}{n}\sum_{i=1}^{n}\ell\big(f(\boldsymbol{x}^{(i)}), \boldsymbol{y}^{(i)}\big),$$

for some surrogate loss $\ell : [0,1]^m \times \{0,1\}^m \to [0,1]$ (e.g. BCE, focal, IoU, contrastive loss). The corresponding population (true) risk is

$$L(f) := \mathbb{E}\big[\ell(f(\boldsymbol{X}), \boldsymbol{Y})\big], \qquad L^{\star} := \inf_{f \in \mathcal{F}} L(f).$$

Remark that in this section we will use the notation $\boldsymbol{X}$ and $\boldsymbol{Y}$ when mass spectra and labels are random variables (both can be random here, unlike Section 4.2 and Appendix G, where only the labels were random variables). Given a trained predictor $f$, we obtain a decision rule by a decoding map

$$\pi_f : \mathcal{X} \to \mathcal{C}(\boldsymbol{x}), \qquad \boldsymbol{x} \mapsto \pi_f(\boldsymbol{x}),$$

which depends on the modeling choice:

- for listwise/contrastive models, $\pi_f(\boldsymbol{x})$ typically selects the candidate with largest score $g_f(\boldsymbol{x}, \mathbf{c})$, cf. (5) to (8);
- for bitwise or vectorwise models, $\pi_f(\boldsymbol{x})$ may select the candidate that maximizes a similarity to $f(\boldsymbol{x})$ (Tanimoto, cosine, etc.).

The induced rule $\pi_f$ can then be evaluated under any utility $U : \mathcal{Y} \times \mathcal{Y} \to [0,1]$ (HR@1, HR@k, Tanimoto, cosine, bitwise accuracy, ...). Recall the definition of a Bayes-optimal decision in Eq. (9), then the *conditional $U$-regret* of a decision rule $\pi$ at $\boldsymbol{x}$ is

$$\mathcal{R}_U(\pi \mid \boldsymbol{x}) := \mathbb{E}\big[U(\boldsymbol{Y}, \boldsymbol{y}_U^{\star}(\boldsymbol{x})) - U(\boldsymbol{Y}, \pi(\boldsymbol{x})) \mid \boldsymbol{x}\big],$$

and the global regret is $\mathcal{R}_U(\pi) := \mathbb{E}_{\boldsymbol{X}}[\mathcal{R}_U(\pi \mid \boldsymbol{X})]$. We also recall the Bayes-level (plug-in) regret of decoding with $V$ when evaluated under $U$:

$$\mathcal{R}_U^{\text{vs }V}(\boldsymbol{x}) := \mathbb{E}\big[U(\boldsymbol{Y}, \boldsymbol{y}_U^{\star}(\boldsymbol{x})) - U(\boldsymbol{Y}, \boldsymbol{y}_V^{\star}(\boldsymbol{x})) \mid \boldsymbol{x}\big],$$

which has been bounded explicitly for several $(U, V)$ in Section 4.1 and Appendix G.4.

**Lemma 2** (Exact decomposition). *For any decision rule $\pi$ and utilities $U, V$,*

$$\mathcal{R}_U(\pi) = \mathbb{E}_{\boldsymbol{X}}\big[\mathcal{R}_U^{\text{vs }V}(\boldsymbol{X})\big] + \mathbb{E}_{\boldsymbol{X}}\big[\Delta_U^{(V)}(\pi \mid \boldsymbol{X})\big], \tag{14}$$

*where*

$$\Delta_U^{(V)}(\pi \mid \boldsymbol{x}) := \mathbb{E}\big[U(\boldsymbol{Y}, \boldsymbol{y}_V^{\star}(\boldsymbol{x})) - U(\boldsymbol{Y}, \pi(\boldsymbol{x})) \mid \boldsymbol{x}\big].$$

*Proof.* For any fixed $\boldsymbol{x}$, add and subtract $U(\boldsymbol{Y}, \boldsymbol{y}_V^{\star}(\boldsymbol{x}))$:

$$\mathcal{R}_U(\pi \mid \boldsymbol{x}) = \mathbb{E}\big[U(\boldsymbol{Y}, \boldsymbol{y}_U^{\star}(\boldsymbol{x})) - U(\boldsymbol{Y}, \pi(\boldsymbol{x})) \mid \boldsymbol{x}\big]$$
$$= \mathbb{E}\big[U(\boldsymbol{Y}, \boldsymbol{y}_U^{\star}(\boldsymbol{x})) - U(\boldsymbol{Y}, \boldsymbol{y}_V^{\star}(\boldsymbol{x})) \mid \boldsymbol{x}\big] + \mathbb{E}\big[U(\boldsymbol{Y}, \boldsymbol{y}_V^{\star}(\boldsymbol{x})) - U(\boldsymbol{Y}, \pi(\boldsymbol{x})) \mid \boldsymbol{x}\big]$$
$$= \mathcal{R}_U^{\text{vs }V}(\boldsymbol{x}) + \Delta_U^{(V)}(\pi \mid \boldsymbol{x}).$$

Taking expectations over $\boldsymbol{X}$ yields (14). $\qquad\square$

The decomposition in Lemma 2 is purely algebraic and does *not* rely on any inequality between $U$ and $V$:

- the first term, $\mathbb{E}[\mathcal{R}_U^{\mathrm{vs}\,V}(\boldsymbol{X})]$, is an intrinsic *Bayes-level mismatch* between decoding with $U$-optimal and $V$-optimal rules;

- the second term, $\mathbb{E}[\Delta_U^{(V)}(\pi \mid \boldsymbol{X})]$, measures how much the learned rule $\pi$ deviates from the $V$-Bayes rule, as seen under the utility $U$. In theory this term can be negative, which happens when the learned rule $\pi$ with surrogate loss based on $V$ is closer to the Bayes-optimal than the $V$-Bayes rule. However, when $\ell$ is a proper or calibrated surrogate for the utility $V$, this term will be positive with very high probability.

Our regret theorems in Section 4.1 and Appendix G.4 provide explicit bounds on the first (Bayes) term for concrete $(U, V)$ pairs (e.g. HR@1 vs. Tanimoto, HR@1 vs. bitwise accuracy). The second term is where generalization and the choice of surrogate loss enter. Numerous papers in statistical learning theory have suggested generalization bounds for the second term. Many of these bounds are independent of the particular utilities $U$ and $V$; they only depend on the complexity of the hypothesis class $\mathcal{F}$ (see e.g. bounds based on empirical Rademacher complexity). Let $\hat{f}_n$ be any empirical risk minimizer in $\mathcal{F}$, i.e.,

$$\widehat{L}_n(\hat{f}_n) = \inf_{f \in \mathcal{F}} \widehat{L}_n(f).$$

Then, classical results in learning theory show that the *excess surrogate risk* $L(\hat{f}_n) - L^\star$ scales with the complexity of $\mathcal{F}$ and decreases as $n$ grows (Bartlett et al., 2006). This is the standard generalization term for the fingerprint predictor.

For concrete surrogate–metric pairs (e.g. BCE for bitwise accuracy, IoU loss for Tanimoto), existing calibration results relate *excess surrogate risk* to *metric-specific regret*. Such calibration inequalities are standard in the statistical learning literature; for instance, logistic/BCE loss is calibrated for $0/1$ accuracy (Bartlett et al., 2006), and similar results hold for top-$k$ probabilities (Yang & Koyejo, 2020). Combining these calibration inequalities with the exact decomposition Eq. 14 yields metric-dependent finite-sample bounds of the form:

$$\mathcal{R}_U(\pi_{\hat{f}_n}) = \underbrace{\text{Bayes mismatch between } U \text{ and } V}_{\text{analyzed in Section 4.1}} + \underbrace{\text{generalization term that shrinks with } n}_{\text{controlled by } L(\hat{f}_n) - L^\star}.$$

We intentionally leave this second part in a general form: the exact constants and functional form depend on $(U, V, \ell)$ and on the decoding $\pi_f$. In summary the second term depends on:

- how well the surrogate-trained predictor $\hat{f}_n$ approximates the $V$-Bayes decision rule, and
- how changes in decisions (from $\boldsymbol{y}_V^\star(\boldsymbol{x})$ to $\pi_{\hat{f}_n}(\boldsymbol{x})$) affect the utility $U$.

