# OpenReview forum: "Small molecule retrieval from tandem mass spectrometry: what are we optimizing for?"
_ICLR.cc/2026/Conference — Submitted to ICLR 2026_

### Official Review · Reviewer_BxDE · 2025-10-17

**Soundness:** 2
**Presentation:** 3
**Contribution:** 2
**Rating:** 4
**Confidence:** 3

**Summary:**

The authors study the task of retrieving a molecular structure from a target database given its tandem mass spectrum. Specifically, they provide empirical and theoretical analysis of commonly used loss functions for molecular fingerprint prediction from a tandem mass spectrum.

**Strengths:**

- Identification of a “paradoxical” trade-off between fingerprint prediction accuracy and downstream retrieval accuracy.
- Novel theoretical results, including regret analyses for common loss functions.
- The paper is generally well structured and clear.

**Weaknesses:**

### Major concerns

- The main limitation is the lack of practical implications. Can the authors demonstrate how their theoretical findings could, for example, guide the design of improved loss functions or inform fingerprint selection? Could these insights improve state-of-the-art results on MassSpecGym? In Table 3, the current Contrastive Emb-Cos results substantially underperform MIST on retrieval with mass-based candidate sets.
- While the paper provides substantial new theory, it is not clear why the Bayes-optimal decision framework is the most appropriate lens here. An alternative path to understanding the paradox would be, for example, an empirical analysis of the per-bit distributions in the ground-truth fingerprints. Some bits may simply be more informative for retrieval, and bitwise or vectorwise losses may lack the capacity to reflect this by weighting every bit equally. Please better justify and explain the choice of the Bayes-optimal framework and significance of the associated theoretical findings.
- The scope of the paper is quite limited considering that fingerprint prediction is one of possible approaches to predict molecular structures from tandem mass spectra.

### Minor concerns

- Although not elaborated there, Goldman et al. previously observed a similar paradox (Nature Machine Intelligence, 2023, https://www.nature.com/articles/s42256-023-00708-3): MIST outperforms SIRIUS in fingerprint prediction accuracy but underperforms SIRIUS in retrieval accuracy (see Fig. 2e and Fig. 3b).
- Line 406: Please clarify what exactly rule V denotes. Is it equivalent to a loss function?
- Lines 367–368: Please justify this statement, for example by adding a reference.

**Questions:**

- Lines 864–866: Could the authors clarify the hypothesis that mass-based candidate sets are more challenging than formula-based candidates. Intuitively, I would expect the opposite, since molecules sharing the same formula should on average be more similar than those sharing only the same mass.

---

> ### Author Response · Authors · 2025-11-23
>
> We begin with a brief common statement for all reviewers. First of all, thank you to all the reviewers for considering our manuscript and providing feedback. We believe the reviews raise insightful points of discussion. Below, we address each concern bullet-by-bullet. Please take into account that we have referenced upcoming changes to our manuscript in multiple places, but that no updated version is available yet. We hope to upload an updated version of the manuscript by the weekend of 29-30th November.
>
> ### Reviewer specific rebuttal
>
> - Regarding practical implications of the theory, MIST performance and fingerprint selection.
>
> For our answer in relation to implications of theory and fingerprint selection: please refer to our response to Reviewer 1 (Lz3p) under the point "Regarding choice of fingerprint and generality. In addition, for our answer w.r.t. MIST performance: please refer to our response to Reviewer 1 (Lz3p) under "Regarding simple model architecture".
>
>
> - Regarding the choice of theoretical framework.
>
> We chose a Bayes-optimal / regret framework because it provides a model and data independent fundamental lens into why and when a similarity objective will or will not align with retrieval performance. We believe a purely descriptive, per-bit empirical analysis cannot achieve this on its own due to the fact that such an analysis does not abstract away from the currently available data. Concretely, the Bayes decision rule is the unique rule that minimizes expected loss given the data distribution, and regret measures the gap to this unattainable ideal: this lets us quantify inherent limits (distributional trade-offs) that are independent of the exact modeling and data setup. In this sense, we believe our approach is uniquely the most appropriate way to study the observed trade off. Regarding per-bit distributions: it is important to note that the Bayes-optimal decision framework already captures per-bit informativeness. The easiest place to note this in Section 4.1 (Bitwise loss functions). There, the Bayes-optimal decision for bitwise accuracy is already defined through posterior bit distributions.
>
> - Regarding the limited scope:
>
> Although fingerprint prediction is one of several possible routes for MS/MS-based structure inference, it remains the only approach that is reliably accurate, scalable, and widely deployed. De novo models still struggle with accuracy (see MassSpecGym benchmark results and the DiffMS model by Bohde et al.), and we believe spectrum simulation for database search is not yet mature enough to serve as a general replacement.
> In contrast, we believe fingerprint prediction is (and likely will remain for quite some time) the most accurate, scalable, and widely adopted paradigm for MS/MS-based structure prediction. Focusing our theoretical analysis on this setting therefore maximizes relevance to current (and future) metabolomics practice.
>
> - Regarding the fact this is also observed in MIST:
>
> The reviewer is absolutely correct this trade-off is also observed in their work. In a revised version we will upload soon, we will add a statement that our trade-off is also reflected in prior work and cite MIST.
>
> - Regarding what exactly rule V denotes.
>
> Rule V denotes a decision rule - a method that does inference. In our analysis, V is always a method that performs optimal inference under some other metric, which is why we substitute V with names of different metrics/losses. Our notation $R\_{U}^{\mathrm{vs}\\,V} $, e.g. $R_\text{Tan}^{\mathrm{vs}\\,\text{Bit}}$ should be understood as regret of a method optimal under Bit similarity when evaluating with Tanimoto similarity.
>
> - Regarding the statement at Line 367–368:
>
> We will add a reference justifying this claim.
>
> - Regarding the hypothesis that mass-based candidate sets are more challenging than formula-based candidates
>
> The statements there should be interpreted not in light of a candidate set being challenging “overall”, more so as the mass-based candidate containing “more” challenging examples to learn from. While the average similarity of any retrieved candidate to the true molecule will be higher if selecting based on formula as opposed to on mass, one needs to also consider how many candidates may be retrieved in both settings, and how retrieving more candidates opens the door to a higher probability that one of those will be very similar. Since we see in Table 5 that training on equal-mass candidates delivers superior performance, even when evaluated in equal-formula settings, we hypothesize that equal-mass candidates contain more “hard” candidates that results in a better discriminative retrieval model.

---

### Official Review · Reviewer_QbNE · 2025-10-19

**Soundness:** 3
**Presentation:** 3
**Contribution:** 3
**Rating:** 6
**Confidence:** 3

**Summary:**

This paper addresses the task of identifying chemical compounds from mass spectra. The approach is using fingerprint prediction,
which involves two steps: 1) predicting fingerprints from spectra, and 2) using the predicted fingerprints to search for candidate compounds in chemical databases.

Main contributions: the paper investigate various loss functions for training fingerprint prediction models and evaluate their performance in terms of both fingerprint prediction accuracy and retrieval performance (i.e. hit rate). Interestingly, the study  empirically shows that achieving higher fingerprint accuracy do not necessarily results in better retrieval performance. Furthermore, the paper provides a theoretical analysis based on Bayes-optimal decision and regret bound to explain these findings.

**Strengths:**

1) the paper is well written and structured.
2) the empirical findings are both interesting and quite surprising - while improving fingerprint accuracy is often considered beneficial, the results demonstrate that it does not necessarily lead to better retrieval performance, which is the ultimate goal in the compound identification task.
3) the theoretical analysis provides an insightful explanation for this observed phenomenon.

**Weaknesses:**

1) the regret analysis in Theorem 2 and 3 is stated in terms of expected loss for Bayes-optimal decision rules, while fingerprint predictors are trained by minimizing. the empirical loss. The theory would be significantly strengthened by connecting the two: introduce the generalization error of the fingerprint predictor into the regret bounds so that regret of the decision rules is expressed as the sum of (i) regret under the empirical losses, and (ii) a provable generalization term.
2) minor issues / typos:
- Eq (1) (line 166): y_{k}-> y^{(i)}_{k}
- Line 1134: equation 10 -> equation (10) or Eq. (10).

**Questions:**

See the weaknesses.

**Details Of Ethics Concerns:**

I have no ethics concerns.

---

> ### Author Response · Authors · 2025-11-23
>
> We begin with a brief common statement for all reviewers. First of all, thank you to all the reviewers for considering our manuscript and providing feedback. We believe the reviews raise insightful points of discussion. Below, we address each concern bullet-by-bullet. Please take into account that we have referenced upcoming changes to our manuscript in multiple places, but that no updated version is available yet. We hope to upload an updated version of the manuscript by the weekend of 29-30th November.
>
> ## Reviewer specific rebuttal
>
> - On expanding the regret bounds with generalization errors
>
> The regret analysis in Section 4.1 is carried out at the Bayes level: we compare decision rules that are optimal under different utilities U and V (e.g. HR@1, Tanimoto, cosine, bitwise accuracy). In practice, however, fingerprint predictors are indeed trained on a finite sample by minimizing an empirical loss. The resulting decision rule therefore deviates from the Bayes-optimal rule. In a new appendix that we will upload soon as part of the revised version of the paper, we discuss how the regret can be decomposed into two phenomena: (1) a purely intrinsic Bayes-level mismatch between using U -optimal or V -optimal decisions
> (captured by the regret bounds in Section 4.1 and Appendix F.4), and (2) a finite-sample generalization component arising from the estimation error of the fingerprint predictor trained by empirical risk minimization (ERM). Formally, one can obtain an exact decomposition of the U-regret into a Bayes term plus a residual term, and we will discuss how the residual can be controlled via standard learning-theoretic bounds on the surrogate loss.
>
> - On the typos
>
> We want to thank the reviewer for their careful reading of our text. In the revised manuscript, these errors have been corrected.

---

> > ### Comment · Reviewer_QbNE · 2025-11-27
> >
> > I thank the authors for clarification. I'd like to keep the score unchanged.

---

### Official Review · Reviewer_UqR9 · 2025-10-31

**Soundness:** 3
**Presentation:** 3
**Contribution:** 2
**Rating:** 4
**Confidence:** 4

**Summary:**

A key problem when analyzing massspec data is to map from an observed spectrum to a molecule. One way to do this is to map from the spectrum to a fixed-length representation for a molecule and then perform database search to find candidate molecules with similar representations. This paper studies an important detail of this approach: the loss function used when training the spectrum -> representation function. Different loss functions can provide very different performance on the downstream molecule retrieval task, even when the loss function rewards accurate prediction of the representation.

This tradeoff is demonstrated empirically on a recent massspec benchmarking task and then analyzed theoretically.

**Strengths:**

The paper points out some key flaws in the algorithmic approach taken by prior work and offers some interesting observations, both empirical and theoretical, for how to navigate them.

The massspec problem is challenging and has broad applications if new methods are accurate at it.

**Weaknesses:**

It was hard for me to understand what the central contribution of this paper was. It targets an interesting application, but does not provide a new modeling technique or achieve SOTA performance on the application. It provides an interesting observation about the impact of different loss functions on eval metrics. Then, it does some theoretical work to show why this dependence on the loss function may exist. If reviewing as an applied paper, the novelty seems limited because the task and models are established. If reviewing as a theory paper, I would need to see more exposition on the relationship between these theory results and other work (e.g., related regret bounds). The empirical observation and theory, that optimizing for a loss function that isn't a good surrogate for the downstream metric can lead to bad performance on the downstream metric, is not particularly surprising.

It was hard for me to understand the MassspecGym results, since the relationship to prior work was unclear. See question below.

**Questions:**

Based on the text in sec 2.2, I would have expected that there was a row in Table 1 in the 'this study' section that is the same setup as MIST (the Goldman paper). However, the MIST numbers seem to be qualitatively different from anything in the 'this study' section. What accounts for the difference?

Can you explain what is novel in the theoretical analysis (besides the particular application to massspec)? Are there results here that would be of general interest to the ICLR community that haven't appeared in prior work?

A central part of the message of the paper is that 'better' fingerprint prediction does not lead to better molecule identification. To me, it's unclear, what better prediction of a fingerprint is, given that this is just some surrogate representation for a molecule, and we don't even know what it would mean to be 'better' (but not exactly perfect) at predicting a molecule. To what extent are your results contingent on the choice of molecular fingerprints? What if you had used, for example, embeddings from a graph neural network encoder? I'm not suggesting that you run such an experiment (which would be complex to set up), but I'd like some thoughts on the robustness of your claims to the choice of that particular molecular representation.

---

> ### Author Response · Authors · 2025-11-23
>
> We begin with a brief common statement for all reviewers. First of all, thank you to all the reviewers for considering our manuscript and providing feedback. We believe the reviews raise insightful points of discussion. Below, we address each concern bullet-by-bullet. Please take into account that we have referenced upcoming changes to our manuscript in multiple places, but that no updated version is available yet. We hope to upload an updated version of the manuscript by the weekend of 29-30th November.
>
> ### Reviewer specific rebuttal
>
> - Regarding central contribution:
>
> The reviewer raises a very fair point that our paper has a foot in both the applied and theoretical camps, and therefore may come off as unconvincing in both respects. We believe however that theoretical insight is often missing in applied domains, limiting deeper understanding. In our case, theory explains and generalizes an empirical trade-off and yields practical implications: minimizing regret ensures that better similarity predictions translate into better retrieval. Our results show that regret grows with (1) the maximum similarity of any negative fingerprint to the true one and (2) the spread of similarities among negatives (Lines 470 and 429), offering concrete guidance for choosing fingerprint representations. Furthermore, we would like to stress that our theoretical results are novel for the machine learning field, because the metabolomics setting is a bit different than problem settings that are typically studied in ML. Fingerprint prediction is a multi-label classification problem, but metrics like IoU and cosine similarity are rarely analyzed in multi-label classification. On the other hand, HR@k is an information retrieval metric, but connections with multi-label loss functions are rarely made in information retrieval. Thus, the relationships we analyze and the accompanying theorems are new to the ML literature. The theorems that we propose are all novel, and the proofs can be found in some of the appendices.
>
> - Regarding the experimental setup and MIST:
>
> From Line 253 onward, we detail our architecture, but we agree the connection to prior work such as MIST could be clearer. In a revised manuscript, we will add a dedicated appendix on related work in fingerprint prediction, including a clearer comparison to MIST. We chose a simple MLP over binned spectra rather than MIST’s chemical formula transformer because strong non-transformer baselines remain competitive: MIST itself reports CSI:FingerID (an SVM ensemble) performs similarly in many cases. As such, our MLP is a desirable choice since it is easy to reproduce, easy to tune, and avoids confounding our analysis with architectural side effects.. Although pushing state-of-the-art performance was not our goal, we note that (1) in the formula-based candidate setting our models outperform MIST, and (2) in the mass-based setting MIST suffers from data leakage by using the ground-truth formula, violating the equal-mass assumption and inflating results. Only the equal-formula setting compares models fairly. We can retroactively estimate equal-mass performance when the formula is known by filtering candidates at inference; for our contrastive “Fp-Cos” model the HR@20 jumps from 43.81% to 84.03%, far above MIST’s reported numbers. In the revised manuscript, we will mark MIST with an asterisk and clearly note this leakage, following the MassSpecGym comparison guidelines (see discussion in issue #52 on their GitHub).
>
> - Regarding the novelty in our theoretical analysis:
>
> All theorems displayed in the main text from Section 4 are novel and have (to our knowledge) not been formulated elsewhere. The most central contribution w.r.t. the observed trade off in MS/MS data is the regret analysis between HR@1 maximizers and bitvector similarity maximizers. There are many other application domains where the setting is similar. For example, in deep hashing, the goal is to predict a binary multi-label hash vector for a single input. The final predicted hash vectors are then used to retrieve from a hash vector database. As such, we feel it will be of broader interest when presented to the ICLR community. We admit we have not expressed the novelty of our theory clearly enough in the current manuscript and will emphasize this better where necessary in the revised version.
>
> - Regarding choice of fingerprint and what it means to predict “better” fingerprints
>
> For our answer to this point: please refer to our response to Reviewer 1 (Lz3p) under the point "Regarding choice of fingerprint and generality:"
>
> To expand a bit on using molecular graph embeddings: this would be an interesting future direction for contrastive models. However, we are not aware of any studies that directly predict the continuous graph embeddings themselves with similarity-based losses. As such, while our theory could be expanded to fit this scenario, we feel that it is less practically relevant within the existing literature.

---

> > ### Comment · Reviewer_UqR9 · 2025-11-25
> >
> > Thanks for the feedback. I have raised my score to weak accept. While I understand that the theoretical analysis is novel, they concern a fairly narrow set of applications and the result is not particularly surprising or actionable. It is well understood in machine learning that minimizing some differentiable surrogate loss may not provide a good model in terms of some downstream performance metric. The mismatch between the training loss in downstream metric are clear.

---

> > > ### Author Response · Authors · 2025-11-27
> > >
> > > Thank you very much for taking the time to re-evaluate our work and for raising your score. We appreciate your engagement with the manuscript.
> > >
> > > We would like to offer one final brief clarification, since the framing in your final comment may suggest that our contribution centers on the well-known fact that surrogate losses can diverge from downstream metrics.
> > > Our intention is not to highlight the generic surrogate--metric mismatch, which is indeed well understood in ML.
> > > Rather, our Bayes-risk (and regret) analyses show how optimizing for one metric delivers suboptimal decisions w.r.t. another metric (irregardless of surrogate loss).
> > > In metabolomics, many methods that predict fingerprints still primarily optimize for fingerprint similarity, all the while silently assuming this will enhance retrieval.
> > > Our manuscript not only theoretically proves this to be suboptimal, in practice, we reveal that the best fingerprint predicting models are the worst at retrieval.

---

### Official Review · Reviewer_nRim · 2025-11-01

**Soundness:** 2
**Presentation:** 3
**Contribution:** 2
**Rating:** 4
**Confidence:** 3

**Summary:**

The paper analyzes different loss functions for molecule retrieval from tandem mass spectra via fingerprint prediction. While the paper is well written and comprehensive, its practical significance is unclear, and the experimental setup requires revision.

**Strengths:**

- The paper is well written and easy to follow.
- Standard datasets are used for analysis.

**Weaknesses:**

- The experimental design appears to be flawed. The paper currently demonstrates (e.g., Fig. 1) that using supervised losses for fingerprint prediction (e.g., BCE) results in better fingerprint prediction, while using supervised losses for retrieval (e.g., contrastive loss) results in better retrieval performance. This is an expected outcome and does not adequately support the claim that there is a “fundamental trade-off between the two objectives.” This claim could be better supported by, for example, analyzing how the performance of both tasks evolves when training for one of the objectives. Such an analysis could reveal, for instance, that training for fingerprint prediction improves retrieval up to a certain point, after which it begins to decrease.
- The practical significance of the paper is not clear. It seems that the main message is that accurate fingerprint prediction and retrieval cannot be achieved simultaneously. However, it is unclear whether this limitation is meaningful in practice. The paper does not propose any measures to address this issue or suggest, for example, that retrieval via fingerprint prediction is suboptimal and that better retrieval performance might be achieved by bypassing fingerprint prediction.

**Questions:**

1. How does the retrieval performance evolve over training time when optimizing for fingerprint prediction (e.g., using BCE loss)?

---

> ### Author Response · Authors · 2025-11-23
>
> We would like to start our rebuttal to each reviewer with a common opening statement. First of all, thank you to all the reviewers for considering our manuscript and providing feedback. We believe the reviews raise insightful points of discussion. In the following, we address each raised point of concern to the reviewer bulletpoint-wise. Please take into account that we have referenced upcoming changes to our manuscript in multiple places, but that no updated version is available yet. We hope to upload an updated version of the manuscript by the weekend of 29-30th November.
>
> ### Reviewer specific rebuttal
>
> - Regarding supporting the claim that there is fundamental trade-off between the two objectives
>
> We agree that our experiments by themselves do not adequately support the claim that there is a fundamental trade-off. Rather, this statement stems from the results in Section 4, where we present novel theory demonstrating that optimal classifier for one objective can perform badly on the other and vice versa. We do this through the lens of the Bayes-optimal decisions for each objective, and how they differ (i.e., Regret). As such, our experiments are just a demonstration of what our theory proves to be true more generally.
>
> As the reviewer suggests, we believe analyzing how performance evolves over training time is valuable and could be complementary to our current theory. We have these graphs available, since we logged these metrics on validation sets during our training runs, we will include them in an Appendix in the revised manuscript.
> However, we can already say that these graphs show that performance over training time is dominated by overfitting dynamics. For example, the contrastive loss specifically optimizes for retrieval, and one can see that overfitting on this metric starts even after a few thousand training steps. The same is visible for the IoU Loss w.r.t. Tanimoto similarity. On the other hand, metrics that the loss functions do not specifically optimize for steadily increase and then plateau. It is important to note that our experimental setup accounts for this: for each model, evaluation for a metric is performed using the checkpoint at which time that model was optimal for that metric. These plots show that validation performance over training time is determined more by overfitting rather than a notion of how optimal one loss is for a metric. As such, we are not convinced that performing such an analysis in the data-restricted regime of metabolomics is a more appropriate lens to explain the observed trade-off.
>
> - Regarding unclear practical significance, addressing the trade-off, and bypassing fingerprint prediction.
>
> We believe our paper holds significant practical implications. Fingerprint prediction is the main paradigm for chemical database retrieval in this field. Many researchers often spend considerable efforts in improving the predicted fingerprints of their model (using losses and metrics to optimize, for example, the Tanimoto similarity), all the while expecting this to improve retrieval results. Our paper is the first to demonstrate empirically that optimizing for similarity is suboptimal w.r.t. retrieval. Further, our theoretical section thoroughly explains how this problem arises and presents a fundamental trade-off. Finally, our theory surfaces theoretically principled guidelines for optimizing retrieval more appropriately: our paper explicitly suggests that fingerprint prediction is suboptimal and that researchers are advised to use contrastive losses (i.e., to bypass fingerprint prediction) if their goal is retrieval (Line 469-471). Further, our theory has practical implications regarding fingerprint selection: minimizing regret is favorable, as in that case, improving fingerprint predictions is expected to improve retrieval. Practically, we show that the regret increases with (1) the maximum similarity of a negative fingerprint to the true one and (2) the difference between the maximum and minimum similarity of a negative fingerprint to the true one (see Line 470 and Line 429). These results serve as practical guidelines for addressing the observed trade off through selecting specific types of fingerprint (e.g., Morgan fingerprints with larger radii).
>
> We do admit that the practical implications of our theory have not been made sufficiently clear in the current version of the manuscript. To address this, in the coming weeks, we will submit a revised manuscript that expands the discussion with parts of the above text.

---

### Official Review · Reviewer_Lz3p · 2025-11-04

**Soundness:** 3
**Presentation:** 3
**Contribution:** 3
**Rating:** 6
**Confidence:** 2

**Summary:**

The paper investigates how different loss functions used to train molecular fingerprint predictors influence the trade-off between fingerprint similarity and molecular retrieval performance in LC-MS/MS analysis. Through empirical evaluation and Bayes-risk-based theoretical analysis, the authors show that improving fingerprint accuracy typically worsens retrieval performance, and vice versa. The work identifies a Pareto trade-off and provides regret bounds explaining why no single loss function can optimize both objectives simultaneously.

**Strengths:**

1. Clearly identifies and explains a practically important but previously overlooked trade-off in molecular retrieval.

2. Provides a unified theoretical framework that aligns well with empirical results.

3. Uses a standardized benchmark with appropriate splitting to ensure fair evaluation.

4. Systematic comparison across a wide range of commonly used loss functions.

**Weaknesses:**

1. Choice of Fingerprint Representation May Limit Generality

The study uses only Morgan fingerprints (radius=2, 4096 bits) as the molecular representation throughout all experiments. While this is a widely used fingerprint type, I am not sure whether the observed trade-off between fingerprint similarity and retrieval performance would still hold for other representations, such as MACCS keys, ECFP with different radii, topological torsions, or even learned neural fingerprints. Since different fingerprint types encode structural information with different sparsity patterns and semantic biases, I am uncertain whether the conclusions would generalize. Therefore, the scope of applicability may be narrower than it initially appears.

2. Model Architecture Might Be Relatively Simple

The fingerprint prediction model is based on a relatively straightforward MLP over binned spectra. I am not fully familiar with spectral modeling practices, but recent molecular identification methods often employ more expressive architectures (e.g., Transformers over peak sequences). Because the representational capacity of the backbone influences the learned distributions over fingerprints, I am curious whether the same Pareto trade-off curve would appear if a more expressive model were used. My confidence here is low, but this might be an important consideration for judging how generally the findings apply.

3. Limited Exploration of Potential Hybrid or Multi-Objective Training Approaches

The paper presents the trade-off clearly, but I am not entirely sure whether this trade-off is inescapable in practice, since the study does not evaluate any hybrid losses that deliberately balance retrieval and fingerprint accuracy. For example, a weighted mixture of contrastive and vectorwise objectives, curriculum training, or multi-objective optimization strategies might produce models closer to the middle region of the Pareto frontier. It would be helpful to know whether such strategies were tested and found ineffective, or simply not explored.

**Questions:**

- Have the authors attempted hybrid / weighted / multi-objective losses to intentionally balance retrieval and fingerprint similarity?

If so, what behaviors were observed?

---

> ### Author Response · Authors · 2025-11-23
>
> We would like to start our rebuttal to each reviewer with a common opening statement. First of all, thank you to all the reviewers for considering our manuscript and providing feedback. We believe the reviews raise insightful points of discussion. In the following, we address each raised point of concern to the reviewer bulletpoint-wise. Please take into account that we have referenced upcoming changes to our manuscript in multiple places, but that no updated version is available yet. We hope to upload an updated version of the manuscript by the weekend of 29-30th November.
>
> ### Reviewer specific rebuttal
> - Regarding choice of fingerprint and generality:
>
> We indeed use Morgan (r=2, 4096-bit) fingerprints because they are the de-facto standard in metabolomics retrieval benchmarks.
> However, none of our theoretical results depend on this choice. The theory in Section 4.2 abstracts away from any specific fingerprint type, radius, dimensionality, or similarity metric by formulating regret in terms of general “similarity bands” (see Line 410). Theorem 5 (Line 1170) further shows that agreement between optimal hit@1 and similarity-based decisions can occur, but only under stringent conditions. This means that representations with different bit distributions (e.g., MACCS, other ECFP radii, torsions, learned embeddings) fit naturally into the same framework. Thus, while we empirically demonstrate the trade-off phenomenon with Morgan fingerprints, the scope of the theory is not restricted to them. Moreover, we believe our theory might be used to inform fingerprint selection: minimizing regret is favorable, as in that case, improving fingerprint similarity predictions is expected to improve retrieval. Practically, we show that the regret increases with (1) the maximum similarity of a negative fingerprint to the true one and (2) the difference between the maximum and minimum similarity of a negative fingerprint to the true one (see Line 470 and Line 429). These results serve as practical guidelines in selecting fingerprints when the goal is minimizing regret.
>
> We admit the points above have not been made sufficiently clear in the current version of the manuscript. To address this, in the coming weeks, we will submit a revised manuscript that (1) expands the discussion with parts of the above text, and (2) includes a section with new experiments including different fingerprint types (Morgan with different radii, rdkit fingerprints with different radii, and MAP4 fingerprints), showing how the theoretical results connect back to practical implications.
>
> - Regarding simple model architecture:
>
> While our backbone is a simple MLP over binned spectra, strong non-transformer baselines in mass-spectral identification remain highly competitive. For example, the recent MIST transformer performs only on par with CSI:FingerID (an SVM ensemble), suggesting that architectural complexity offers limited gains given current data sizes. We therefore chose an MLP because it is reproducible, easy to tune, and avoids confounding our analysis with architectural side effects. Since our goal is to study the trade-off rather than push state-of-the-art accuracy, this simpler model is appropriate. Moreover, (1) in the formula-based candidate setting our models outperform the transformer-based MIST, and (2) in the mass-based setting MIST exhibits data leakage because it uses the ground-truth molecular formula, which violates the equal-mass assumption and inflates its scores. Only the equal-formula setting compares models fairly. We can retroactively estimate equal-mass performance when the formula is known by filtering candidates at inference; for our contrastive “Fp-Cos” model the HR@20 jumps from 43.81% to 84.03%, far above MIST’s reported numbers. In the revised manuscript, we will mark MIST with an asterisk and clearly note this leakage, following the MassSpecGym comparison guidelines (see discussion in issue #52 on their GitHub).
>
> - Regarding testing multi-objective losses
>
> While it is true that we did not deliberately try to produce models that occupy different points along the trade-off Pareto curve, it is important to realize that our theorems are written in terms of how optimal predictions differ for both metrics of interest. As such, the trade-off is fundamental: to perform optimally for one metric inescapably means to be suboptimal for the other. The tested loss functions naturally occupy some point on the Pareto frontier.
> We are convinced that multi-hybrid losses will not produce surprising results: they will produce models performing somewhere in between, perhaps similarly to the performance obtained with BCE loss and FL In Figure 1.
> However, it remains interesting to see whether such hybrids lie above or below the currently observed frontier. To address this, we will submit a revised manuscript including experiments that train with combinations of IoU and contrastive losses across a range of weighting schemes.

---

### Author Response · Authors · 2025-12-01
**Summarizing comment**

We have now uploaded a revised version of the manuscript in which we believe all of the reviewers' concerns are addressed (see individual responses to reviewers). To make it easier for the AC to reach a decision, we want to take the opportunity to summarize the main strengths and weaknesses as they have been pointed out by the reviewers, as well as how we addressed the weaknesses.
### Strengths
- Our paper tackles an important and challenging problem in computational metabolomics (Lz3p, UqR9)
- We employ a standardized benchmark for our experimental analysis (Lz3p, nRim)
- The trade-off we empirically identify has been described by reviewers as interesting, surprising, and “paradoxical” (QbNE, BxDE, BxDE). We provide substantial new theory that explains this observation in a unifying and insightful way (UqR9, Lz3p, QbNE, BxDE).
- The paper is well written, clear, and easy to follow (nRim, QbNE, BxDE)
### Weaknesses
Here, we point out some of the limitations that were recurrently cited by reviewers. For an overall list of weaknesses, please refer to the individual reviewers’ comments.

- “Unclear practical implications.” (nRim, UqR9, BxDE)

Reviewers consistently cited that the practical implications of our novel theoretical results were not clear to them. In the revised version, we have added an entirely new section (“4.3 Practical implication of our theoretical analysis”) that illustrates how our theoretical results may inform fingerprint selection toward minimizing the observed trade-off. Our novel experiments confirm our theory and surface MAP4 fingerprints as a promising complementary or alternative fingerprint target. Details are in a new Appendix (H).

- “Limited scope and generality of using Morgan (r=2) fingerprints.” (Lz3p, BxDE)

While our original manuscript indeed used Morgan (r=2) fingerprints because they are the de facto standard in metabolomics retrieval benchmarks, our theory abstracts away from the specific choice of fingerprints. In fact, to put it even stronger, as mentioned above, our theory can inform fingerprint selection. We have now showcased this in Section 4.3, which expands the scope of our experimental results to other fingerprint types.

- “Experiments were performed using a simple architecture (Lz3p), no consistent state-of-the-art results are obtained.” (UqR9)

The inconsistent state-of-the-art advantages over MIST can be explained by the fact that there are data leakage concerns with using chemical formulae in the MassSpecGym benchmark. We have indicated this more clearly in the revised version. In light of this, our well-tuned MLP “baselines” consistently outperform the previous state-of-the-art.


- “Unsurprising core result.” (nRim, UqR9)

We acknowledge that, when one takes a step back, the fact the trade-off exists is perhaps unsurprising. We do believe, however, that its scale is unexpectedly large: the models that achieve the best fingerprint prediction are the worst at retrieval by a wide margin. Our work is the first to reveal and formalize this phenomenon systematically. Because many approaches in this field still optimize fingerprint similarity under the implicit assumption that it improves retrieval, we believe highlighting this disconnect is an important and timely contribution.

---

### Meta-Review · Area_Chair_oY34 · 2026-01-08

**Summary:**

This paper studies the task of retrieving chemical compounds from a molecular database on the basis of LC-MS/MS data. In particular, this paper investigates how different loss functions used in training affect performance of prediction models. Furthermore, this paper provides a regret analysis, where the regret is defined as the difference between the expected utility/similarity of the best decision under the same utility/similarity and that of the best decision under a different utility/similarity.

**Reviewers' concern:**
- Unclear central contribution. (UqR9)
- Practical implications are unclear. (nRim, UqR9, BxDE)
- Limited generality due to use of a specific fingerprint representation. (Lz3p, BxDE)
- Simple architecture of model. (Lz3p)
- Apparent inconsistency in MIST results. (UqR9)
- Trade-off is not surprising. (nRim, UqR9)

**Additional points:**
- Equations (1), (2): The summand should be enclosed by a pair of parenthesis.
- Lines 178, 268, 269: I did not understand what "$\pm$" means.
- The indicator function is denoted by $𝕀(\cdot)$ in lines 150-152, by $[\cdot]$ in line 358, and ${\bf 1}\\\{\cdot\\\}$ in lines 1090-1092. A consistent notation should be adopted.
- Line 312: The expression "HR@20" appears here for the first time. This abbreviation should be introduced explicitly.
- Line 318: irregardless → regardless
- Lines 320-321: An exception to this rule (are → is)
- Line 323: negative candidate(s) sets
- Line 1011: The quantity $s(\hat{y})$ is defined but is never used in the proof.
- Line 1112: The symbol $u_i$ has already been used in Theorem 1 to denote the quantity $\sum_y\frac{y_i p(y\mid x)}{||y||}$, so that reusing it to denote a different quantity is confusing and should thus be avoided.
- Lines 1186-1187: The same formula appears twice.
- Line 1197: for all $t\not=i_\star$ → for all $t\not\in\\\{i_\star,i_{sim}\\\}$
- Proof of Theorem 3: Relation of $\mathcal{R}_{HR@1}^{vs\\\,sim}$ with quantities dealt with in the proof is not stated explicitly, making the proof incomplete.

**Reviewer Concerns:**

- Unclear central contribution. (UqR9) → I think that the central contribution is still not clear after the revision. In particular, it seems that the theory should be applicable well beyond the task of molecular retrieval from mass spectrometry data. How general the empirical observations, such as the mentioned trade-off, and the theoretical results are, as well as what aspects are specific to the task of molecular retrieval, is still unclear. Whether the specific task of molecular retrieval is chosen merely for demonstration purpose or as the main target application is not clear either.
- Practical implications are unclear. (nRim, UqR9, BxDE) → The revised paper discusses practical implications of the theoretical results in the newly added Section 4.3.
- Limited generality due to use of a specific fingerprint representation. (Lz3p, BxDE) → The revised manuscript discusses fingerprint selection in the newly added Appendix H.
- Simple architecture of model. (Lz3p) → Reasoning on the use of simple model is provided in the author response.
- Apparent inconsistency in MIST results. (UqR9) → This is explained in the author response as being due to possible data leakage.
- Trade-off is not surprising. (nRim, UqR9) → I think that this paper puts too strong emphasis on the observed trade-off as being fundamental.

Because of what are listed above, I think that this paper would benefit from a thorough reorganization clarifying main contributions, including what practical implications are obtained.

**Reviewer Scores:**

Initial score/confidence of Reviewers Lz3p and QbNE were 6/2 and 6/3, and they would not have changed their evaluation after the rebuttal. Reviewer UqR9 initially rated 4/4, but (s)he wrote to raise the score to weak accept. Evaluation of Reviewers nRim (4/3) and BxDE (4/3) would have remained.

---

### Decision · Program_Chairs · 2026-01-26

Reject